# Solubility and Solution-phase Chemistry of Isocyanic Acid, Methyl Isocyanate, and Cyanogen Halides

James M. Roberts[1], and Yong Liu[2]

*1. NOAA/ESRL Chemical Sciences Division, Boulder, Colorado, 80305*
*2. Department of Chemistry, University of Colorado, Denver, Denver, Colorado, 80217*

**Abstract**

Condensed phase uptake and reaction are import atmospheric removal processes for reduced nitrogen species, isocyanic acid (HNCO), methyl isocyanate ($CH_3NCO$) and cyanogen halides (XCN, X =Cl, Br, I), yet many of the fundamental quantities that govern this chemistry have not been measured or are understudied. These nitrogen species are of emerging interest in the atmosphere as they have either biomass burning sources, i.e. HNCO and $CH_3NCO$, or like the XCN species, have the potential to be a significant condensed-phase source of $NCO^-$ and therefore HNCO. Solubilities and first-order reaction rate of these species were measured for a variety of solutions using a bubble flow reactor method with total reactive nitrogen ($N_r$) detection. The aqueous solubility of HNCO was measured as a function of pH, and had an intrinsic Henry's law solubility of 20 ($\pm 2$) M/atm, and a $K_a$ of 2.0 ($\pm 0.3$) $\times 10^{-4}$ M ($pK_a = 3.7 \pm 0.1$) at 298K. The temperature dependence of HNCO solubility was very similar to other small nitrogen-containing compounds, such as HCN, acetonitrile ($CH_3CN$), and nitromethane, and the dependence on salt concentration exhibited the "salting out" phenomenon. The rate constant of reaction of HNCO with 0.45 M $NH_4$, as $NH_4Cl$, was measured at pH=3, and found to be 1.2 ($\pm 0.1$) $\times 10^{-3}$ $M^{-1}s^{-1}$, faster than the rate that would be estimated from rate measurements at much higher pHs. The solubilities of HNCO in the non-polar solvents n-octanol (n-$C_8H_{17}OH$) and tridecane ($C_{13}H_{28}$) were found to be higher than aqueous solution for n-octanol (87 $\pm 9$ M/atm at 298K) and much lower than aqueous solution for tridecane (1.7 $\pm 0.17$ M/atm at 298K), features that have implications for multi-phase and membrane transport of HNCO. The first-order loss rate of HNCO in n-octanol was determined to be relatively slow 5.7 ($\pm 1.4$) $\times 10^{-5}s^{-1}$. The aqueous solubility of $CH_3NCO$ was found to be 1.3 ($\pm 0.13$) M/atm independent of pH, and $CH_3NCO$ solubility in n-octanol was also determined at several temperatures and ranged from 4.0 ($\pm 0.5$) M/atm at 298K to 2.8 ($\pm 0.3$) M/atm at 310K. The aqueous hydrolysis of $CH_3NCO$ was observed to be slightly acid-catalyzed, in agreement with literature values, and reactions with n-octanol ranged from 2.5 ($\pm 0.5$) to 5.3 ($\pm 0.7$) $\times 10^{-3}$ $s^{-1}$ from 298 to 310K. The aqueous solubilities of XCN ~~was~~ determined at room temperature and neutral pH were found to increase with halogen atom polarizability from 1.4 ($\pm 0.2$) M/atm for ClCN, 8.2 ($\pm 0.8$) M/atm for BrCN, to 270 ($\pm 54$) M/atm for ICN. Hydrolysis rates, where measurable, were in agreement with literature values. The atmospheric loss rates of HNCO, $CH_3NCO$, and XCN due to heterogeneous processes are estimated from solubilities and reaction rates. Lifetimes of HNCO range from about 1 day against deposition to neutral pH surfaces in the boundary layer, but otherwise can be as long as several months in the mid-troposphere. The loss of $CH_3NCO$ due to aqueous phase processes is estimated to be slower than, or comparable to, the lifetime against OH reaction (3 months). The loss of XCNs due to aqueous uptake are estimated to range from quite slow, lifetime of 2-6 months or more for ClCN, 1 week to 6 months for BrCN, to 1 to 10 days for ICN. These characteristic times are shorter than photolysis lifetimes for ClCN, and BrCN, implying that heterogeneous chemistry will be the controlling factor in their atmospheric removal. In contrast, the photolysis of ICN is estimated to be faster than heterogeneous loss for average mid-latitude conditions.

## I. Introduction

The earth's atmosphere is a highly oxidizing environment in which chemical compounds are typically destroyed through radical pathways. The reduced nitrogen species, isocyanic acid (HNCO) and hydrogen cyanide (HCN), are an exception to this, as they have slow reactions with atmospheric radicals and have primarily condensed-phase sources and sinks (Li et al., 2000; Roberts et al., 2011). Cyanogen halides (XCN, where X = Cl, Br, I) are compounds that are present in the environment, and whose atmospheric chemistry is of emerging interest. XCN compounds likewise have very slow reaction rates with radical species and, with the exception of ICN, very slow photolysis rates in the troposphere (Keller-Rudek et al., 2013). These general classes of reduced nitrogen species, isocyanates (R-NCO), cyanides (RCN), and cyanogen halides (XCN) have potential health impacts that are related to their condensed phase chemistry (Boenig and Chew, 1999; Broughton, 2005; McMaster et al., 2018; Wang et al., 2007). Therefore, information on solubility and reaction rates are needed to understand the atmospheric fate of such compounds and define their impact on human and ecosystem health. Five reduced nitrogen species will be focused on here: isocyanic acid, HNCO, methyl isocyanate, $CH_3NCO$, which are biomass burning products, and cyanogen chloride, ClCN, cyanogen bromide, BrCN, and cyanogen iodide, ICN, which could be condensed-phase sources of cyanate ion ($NCO^-$) and therefore HNCO.

The isocyanate compounds are products of the pyrolysis or combustion of N-containing materials (biomass, polyurethanes) (Blomqvist et al., 2003; Koss et al., 2018) and the two simplest ones, HNCO and $CH_3NCO$, have also been observed in interstellar and cometary media (Goesmann et al., 2015; Halfen et al., 2015). The atmospheric chemistry of HNCO has received considerable attention in the past few years as it has become clear that it is present in ambient air, and could be related to health impacts through specific biochemical pathways (Roberts et al., 2011) involving the reaction of cyanate ion with proteins. There are relatively few observations of HNCO in ambient air, showing "background" mixing ratios that range from 10pptv to over several ppbv depending on the nature of regional sources, and peak mixing ratios approaching a few ppbv, observed in areas impacted by local biomass burning (Chandra and Sinha, 2016; Kumar et al., 2018; Mattila et al., 2018; Roberts et al., 2014; Sarkar et al., 2016; Wentzell et al., 2013; Woodward-Massey et al., 2014; Zhao et al., 2014). The aqueous phase solubility of HNCO was examined by Roberts et al., (Roberts et al., 2011) and Borduas et al., (2016), wherein it was found that HNCO shows behavior typical of weak acids, where the effective Henry's coefficient $H_{eff}$ varies with pH, and so HNCO is only slightly soluble at pHs characteristic of atmospheric aerosol (pH= 2-4) and is quite soluble at physiologic conditions (pH=7.4). Attempts to model the global distribution of HNCO (Young et al., 2012) and the cloud water uptake of HNCO (Barth et al., 2013) used the limited solubility and hydrolysis data available at that time, (Jensen, 1958; Roberts et al., 2011). Several aspects of HNCO solubility remain unknown, such as salt effects on aqueous solubility, and solubility in non-aqueous solvents, a property important for predicting HNCO behavior in biological systems. The pH dependent hydrolysis of HNCO had been studied some time ago (Jensen, 1958), the mechanism for this process involves three separate reactions;

$$HNCO + H_3O^+ \rightarrow NH_4^+ + CO_2 \qquad\qquad (R1)$$

$$HNCO + H_2O \rightarrow NH_3 + CO_2 \qquad\qquad (R2)$$

$$NCO^- + 2H_2O \rightarrow NH_3 + HCO_3^-$$ (R3)
and Borduas et al. (2016), recently re-measured these rates under a wider range of conditions and found their
measurements to be essentially consistent with the previous work at pHs of interest in the atmosphere. Rates of
reaction of HNCO with other compounds in aqueous solution are not as well studied, especially under atmospheric
conditions, e.g. low pH, relatively high ionic strength. Rates of reaction of HNCO/NCO$^-$ with nitrogen bases have
been measured but only at the pK$_a$s of the BH$^+$, which are typically pH 9-10 (Jensen, 1959; Williams and Jencks,
1974a, b). The pKa is defined as the negative Log$_{10}$ of the dissociation constant of an acid, and can be thought of as
the pH at which the acid and its conjugate base (in this case BH+ and B) are at the same concentration.

Methyl isocyanate is most notable for its part in the one of the largest industrial disasters in history, when a

large quantity of CH$_3$NCO was released from a chemical plant and fumigated the city of Bhopal, India. There are
other, more common sources of CH$_3$NCO to the atmosphere including combustion of biomass (Koss et al., 2018)
and N-containing polymers such as polyurethanes and isocyanate foams (Bengtstrom et al., 2016; Garrido et al.,
2017), and cooking (Reyes-Villegas et al., 2018). Recent measurements of CH$_3$NCO in laboratory wildfire studies
have observed mixing ratios up to 10 ppbv or so in fuels characteristic of western North America (Koss et al., 2018).
CH$_3$NCO is also produced in photochemical reactions of methylisothiocyanate (CH$_3$NCS), which is the main
degradation product of the agricultural fungicide metam-sodium (CH$_3$NHCS$_2$Na) (Geddes et al., 1995). In addition,
CH$_3$NCO has been observed in studies of the photooxidation of amides (Barnes et al., 2010; Borduas et al., 2015;
Bunkan et al., 2015) and by extension will be formed in dimethyl amine oxidation. To our knowledge there is only
one reported set of ambient measurements of CH$_3$NCO, conducted near a field where metam-sodium was being used
as a soil fumigant (Woodrow et al., 2014), and the resulting CH$_3$NCO mixing ratios were as high as 1.7 ppbv. The
California Office of Environmental Health Hazard Assessment has placed an inhalation reference exposure level of
0.5 ppbv (1 $\mu$g/m$^3$) on CH$_3$NCO due to its propensity to cause respiratory health effects (California, 2008).

There have been only a few studies of the gas phase loss rates of CH$_3$NCO including reaction with OH

radical, which appears to be slow based on the mostly recent measurements (Lu et al., 2014) (Papanastasiou et al., in
preparation, 2019), reaction with chlorine atoms (Cl) which might be as much as 20% of OH under some
atmospheric conditions (Papanastasiou et al., in preparation, 2019), and UV photolysis which has a negligible
contribution to atmospheric loss (Papanastasiou et al., in preparation, 2019). Thus, heterogeneous uptake might
compete with these gas phase loss processes. The solubility of CH$_3$NCO has not been previously determined
experimentally, but is probably low, <2 M/atm, by analogy to CH$_3$NCS (3.7 M/atm) (Geddes et al., 1995). In
addition, there are no data on the solubility of CH$_3$NCO in non-aqueous solvents. The hydrolysis of CH$_3$NCO is acid
catalyzed, exhibiting the following overall reactions;
$$CH_3NCO + H_2O + H^+ \rightarrow CH_3NH_3^+ + CO_2$$ (R4)
$$CH_3NCO + H_2O \rightarrow CH_3NH_2 + CO_2$$ (R5)
producing methyl amine and carbon dioxide. The rate constants for these reactions are fairly well established (Al-
Rawi and Williams, 1977; Castro et al., 1985).

Cyanogen halides are less well studied as atmospheric species, but have potentially important

environmental sources. Cyanogen chloride was once produced as a chemical warfare agent, however its importance
to the atmosphere is more related to its possible formation in the reaction of active chlorine species (HOCl/OCl-,
chloramines) with N-containing substrates such as amino acids and humic substances (Na and Olson, 2006; Shang et
al., 2000; Yang and Shang, 2004). These reactions are known to be important in systems where chlorination is used
for disinfection such as swimming pools and water treatment (see for example (Afifi and Blatchley III, 2015), and
perhaps indoor surfaces (J. Abbatt, personal communication). We are not aware of any measurements of ClCN in
ambient air. Cyanogen bromide can likewise be formed through reactions of HOBr/OBr- with reduced nitrogen
species, and there are observations of BrCN in bromide-containing waters that have been received chlorine
treatment (see for example (Heller-Grossman et al., 1999). The formation results from the facile reaction of
HOCl/OCl- with bromide to make HOBr/OBr-, which then reacts with nitrogen species in the water. In addition,
there is a natural source of BrCN from at least one strain of marine algae (Vanelslander et al., 2012) that is thought
to be related to allelopathic activity, i.e. secreted to control the growth of competing organisms. This marine algae
source may be responsible for BrCN levels observed in remote atmospheres (J.A. Neuman and P.R. Veres, personal
communication),(NASA, 2019). Cyanogen iodide can also potentially be formed from the chlorination of water or
wastewater because iodide is easily oxidized by HOCl/OCl$^-$, however iodide is usually quite small in concentration,
so the several studies that report total cyanogen halides report ClCN and BrCN but not ICN (Diehl et al., 2000;
Yang and Shang, 2004). There are also biochemical pathways for ICN formation involving several enzymes that are
part of the immune defense system (see for example (Schlorke et al., 2016)), but the extent to which ICN might be
volatilized from those systems is not clear. There are also some observations of ICN in the remote marine
troposphere (J.A. Neuman and P.R. Veres, personal communication), but their origin is currently unclear.

The possible gas phase loss processes of cyanogen halides include reaction with radicals or ozone, and

photolysis. Radical reaction rates (OH, Cl) have not been measured at room temperatures, but are likely to be slow
due to the strength of X-CN bonds (Davis and Okabe, 1968). The UV-visible absorption spectra of all three of these
compounds have been measured (Barts and Halpern, 1989; Felps et al., 1991; Hess and Leone, 1987; Russell et al.,
1987), and indicate a range of photolysis behavior ranging from no tropospheric photolysis of ClCN, to slight
photolysis of BrCN, and faster photolysis of ICN. The rates of photolysis need to be balanced against condensed
phase losses of XCN compounds to obtain a full picture of their atmospheric losses.

The aqueous phase solution chemistry of cyanogen halides is not as well studied as the isocyanates. The

aqueous solubilities of XCN compounds are not known with the exception of ClCN whose solubility is thought to be
fairly low, 0.6 – 0.52 M/atm at 293-298K (Weng et al., 2011; Yaws and Yang, 1992) as reported by (Hilal et al.,
2008). The hydrolysis of XCN compounds are known to be base-catalyzed and so involve the following reactions;

$XCN + H_2O \rightarrow HOCN + H^+ + X^-$           (R6)

$XCN + OH^- \rightarrow HOCN + X^-$           (R7)


with R6 being fairly slow at medium to low pH (Bailey and Bishop, 1973; Gerritsen et al., 1993). The product,
cyanic acid, HOCN, is unstable with respect to HNCO in aqueous solution (Belson and Strachan, 1982);

$HOCN + H^+ \rightarrow HNCO + H^+$                           (R8)


Thus, XCN compounds represent potential intermediates in the condensed-phase formation of HNCO, for which
there is some observational evidence (Zhao et al., 2014). So, in addition to being active halogen species, XCN
compounds represent potential condensed phase source of HNCO in systems where there is halogen activation and
there are reduced nitrogen species present, e.g. wildfire plumes, bio-aerosols and indoor surfaces.

Measurements of solubility and reaction rates will be presented here for HNCO, $CH_3NCO$, and the XCN

species: ClCN, BrCN, and ICN. The aqueous solubility of HNCO was measured as a function of pH in the range pH
2-4, temperature in the range 279-310K, and salt concentration up to 2.5M NaCl. The rate of reaction of HNCO with
$NH_4^+$ was measured at pH3, to examine the importance of this reaction to atmospheric uptake of HNCO. The
solubilities of HNCO in the non-polar solvents n-octanol and tridecane were also measured as a function of
temperature, in the range 298-310K, and the first-order loss rate of HNCO in n-octanol was also determined. The
aqueous solubility of $CH_3NCO$ was measured ~~at several~~ at several pHs pH 2 and 7, and the solubility in n-octanol
was also determined at several temperatures, 298 and 310K. Finally, the aqueous solubility of ClCN, BrCN, and
ICN were determined at room temperature, and at 273.15 K (ClCN, BrCN) and neutral pH, and the solubility and
first loss of these compounds in n-octanol was also determined. These data will be used to estimate atmospheric
lifetimes against aqueous uptake and to assess the relative bioavailability of these compounds.

**II. Methods**

Most of the techniques used for the work presented here have largely been presented elsewhere (Borduas et

al., 2016; Kames and Schurath, 1995; Kish et al., 2013; Roberts, 2005) and will only be briefly summarized here.
The basic principle is that the compound of interest is equilibrated with solution in a bubble flow reactor, and then
removed from the gas-phase and the exponential decay of the signal due to loss of the compound is measured with a
sensitive and selective method. The dependence of decay rates on flow rate-to-liquid volume ratio can then be
related to solubility and first-order loss rate due to reaction in solution. This technique relies on being able to
produce a consistent gas stream of the compound of interest, and being able to selectively detect the compound
exiting the reactor. This method has limitations in that the solubility must be within a certain range, and the first-
order loss rate slow enough that there are measurable amounts of compound exiting the reactor.

A. Preparation of Gas-Phase Standards

The general system used for preparation of gas phase streams of HNCO, $CH_3NCO$, BrCN, and ICN was the

capillary diffusion system described by (Williams et al., 2000) and (Roberts et al., 2010). Isocyanic acid was
produced in a steady stream by heating the trimer, cyanuric acid (Sigma-Aldrich, USA) to 250°C under $N_2$ and
establishing a constant diffusion rate through a short length of capillary tubing (1mm ID x 5cm length). Care was
taken to condition the system for several days before use, by keeping the system under flow and at a minimum of
125°C even when not in active use, to prevent the build-up of unwanted impurities, particularly $NH_3$. Standards in
the range of several ppmv in 40 SCCM could easily be prepared in this way.

The same capillary diffusion cells were used for $CH_3NCO$ preparation, starting with a sample of the pure

liquid (Alinda Chemicals, UK). FTIR analysis of samples of this material were found to contain small amounts of
siloxanes (3% by mole), which probably came from a chloro-silane added as a stabilizer, but no measurable
presence of any other nitrogen compounds. The high volatility of $CH_3NCO$ (BP 38 °C) required that low
concentration solution (1% vol/vol) of $CH_3NCO$ in n-tridecane ($C_{13}H_{28}$) solvent at a temperature of 0°C be used in
the diffusion cell. Under these conditions a 40 SCCM stream resulted in a mixing ratio of 10ppmv. The output of the
source was stable for long periods of time (days) and could be used for the solubility study and calibration of other
instruments. The source was also analyzed by an $H_3O^+$ chemical ionization mass spectrometric system ($H_3O^+$ CIMS)
(Koss et al., 2018; Yuan et al., 2016), which showed that it had no impurities detectable above the 1% (as N) level.

The preparation of a gas phase standard of ClCN is described by Stockwell et al., (2018) and is based on

chemical conversion of an HCN calibration mixture.  It has been known for some time that HCN reacts readily with
active chlorine compounds to yield ClCN (Epstein, 1947), for example:

$HCN + HOCl \rightarrow ClCN + H_2O$                                            (R9)

In fact, this reaction has been used as the basis for measuring HCN in the gas phase by conversion to ClCN with
detection by gas chromatography with electron capture (Valentour et al., 1974). In those systems, Chloramine-T (*N*-
Chloro-*p*-toluenesulfonamide sodium salt, Sigma-Aldrich) has proven useful. The method used in this work
consisted of passing a small stream (5-10 SCCM) of a commercially-prepared 10ppmv gas-phase standard of HCN
in $N_2$ (GASCO, Oldsmar, FL), combined with humidified Zero Air (ZA, 80% RH, 30-50 SCCM) over a bed packed
with glass beads coated with a solution of Chloramine-T.  The glass beads were prepared by coating glass 3 mm OD
beads with a 2 g/100cc solution and packing ~20cc of them in a 12.7mm OD PFA tube and flowing ZA over them
until they appeared dry. The reaction was shown to be essentially 100% (±10%) when conducted in a humidified
atmosphere (RH ≥60%), ($H_3O^+$ CIMS), and FTIR analysis of the gas stream before and after passing through the
chlorination bed. The ClCN source was also checked by measuring the total nitrogen content of the gas stream
before and after the chlorination step, and the resulting signal was found to be 98±1% of the original HCN standard.
This means that the combination of the chlorination reaction and $N_r$ conversion (see below) were at least 98%
efficient.

Preparation of BrCN and ICN gas streams was accomplished with the diffusion cell apparatus using

commercially available samples of BrCN (98% purity, Sigma-Aldrich) and ICN (97% purity ACROS Organics),
that were used without further purification. BrCN is a volatile solid, so was kept on a diffusion cell at 0°C while in
use. ICN is a relatively non-volatile solid and so was placed in a diffusion cell and heated to 80°C while in use.
These resulted in sample streams that were on the order of 250-350 ppbv in 1 SLPM in mixing ratio. Analysis by
iodide ion chemical ionization mass spectrometry {Warneke, 2016 #1366} indicated traces of the molecular halogen
species ($Br_2$, $I_2$), but no other significant N-containing species.


B. Detection of Nitrogen Compounds

The method for detection of the compounds studied in this work relies on high temperature conversion of any N-containing species, except for $N_2$ or $N_2O$, to nitric oxide (NO) and detection of the resulting NO by $O_3$ chemiluminescence (Williams et al., 1998). This technique, which we will refer to as Total Reactive Nitrogen, $N_r$, has been shown to measure a wide range of reduced nitrogen species as well as the more familiar oxides of nitrogen (Hardy and Knarr, 1982; Saylor et al., 2010; Stockwell et al., 2018), provided care is taken to convert any nitrogen dioxide that is formed in the Pt converter back to NO prior to detection (Schwab et al., 2007). In this work, this was accomplished with a solid molybdenum tube operated at between 350 and 450 °C, with the addition of a small amount of pure $H_2$ resulting in a 0.8% mixing ratio in the catalyst flow. The detection system was routinely calibrated with a NO standard (Scott-Marrin, Riverside, CA) and the conversion efficiency was confirmed with a low concentration (10ppmv) HCN standard (GASCO, Oldsmar, FL).  The high conversion efficiencies (≥98%) for HNCO and ClCN were confirmed by other methods as described by Stockwell et al. (2018). The conversion efficiencies for BrCN, and ICN are assumed to be equally high due to the fact the X-CN bond strengths of these compounds are lower than for H-CN and Cl-CN (Davis and Okabe, 1968) and the $CH_3$-NCO bond is weaker than the H-NCO bond (Woo and Liu, 1935) so $CH_3NCO$ should be easily converted by the Nr catalyst. Although readily measured here, a solubility measurement of this kind does not require the determination of the absolute concentration of the analytes, it only requires that the measurement be linear (i.e. constant sensitivity) throughout the range of signals measured.  The NO instrument is linear from the low pptv into the low ppmv range, the chief limitation being the ability to count photon rates above 5MHz. The magnitude of the gas phase sources used and the flow rate of the instrument (1 SLPM) insured that instrument signals did not reach the non-linear range.

The requirement for the detection method to be selective could be an issue with a general method such as $N_r$. In practice, the reactions of the nitrogen species studied here form products that are not volatile under the conditions used in this work, and so do not interfere with the measurement. In aqueous-phase reactions, HNCO produces $NH_3/NH_4^+$, $CH_3NCO$ produces $CH_3NH_2/CH_3NH_3^+$, and XCN compounds produce $HOCN/NCO^-$ all of which are non-volatile in the pH ranges at which those experiments were conducted. The products of the organic-phase reactions are not as well known: tridecane should not react with HNCO, n-octanol will form carbamyl or methyl carbamyl groups with n-octyl substituents which should be non-volatile. Possible reactions of XCN compounds with n-octanol are less well known, particularly in the absence of water in the solution, so those experiments will need to be interpreted with care.

The reactor used for the most of the experiments is a modification of the one described by Roberts (2005), the main modification being a  reduction in volume to 125 cc. Liquid volumes used in the experiments ranged from 20 to 50 cc, and the volumetric flow rates used ranged from 170 to 1070 ambient cc/min. Temperatures were measured using a calibrated mercury thermometer, and in temperatures different than room temperatures were controlled using a water bath with either ice/water, or a temperature control system. The uncertainties in the temperatures were ±0.5 °C.

The bubble flow reactor method relies on the rapid equilibration of a gas stream that contains the analyte of
interest, with solution by means of the creation of small, finely divided bubbles. In the system used here, these
bubbles are created by passing the gas stream through a fine glass frit, situated at the bottom of the glass vessel.  The
main sample flow is passed through the bubbler and into the detector stream to establish a baseline. A small flow of
the analyte is added upstream of the reactor by means of a PFA solenoid valve to start the measurement and the
effluent is monitored until the measured concentration attains equilibrium. At this point, the analyte entering the
reactor is switch off, and the concentration exiting the reactor begins to decay. This decay is due to a combination of
loss of the analyte as it re-equilibrates with the gas stream, and first-order loss in the solution due to reaction. Under
conditions of rapid equilibration, this decay takes the form of a single exponential equation, dependent on the ratio
of flow rate ($\phi$, cm$^3$/s) to liquid volume (V, cm$^3$), the effective Henry's Law solubility $H_{eff}$ (M/atm), and the first-
order loss rate ($k$):

$ln(C_0/C_t) = [\phi/(H_{eff}RTV) + k]t$                                      Eq. (1)


Measurements performed at a series of $\phi$/V should be linear with a slope of the decay rate (d $ln(C_0/C_t)$/d$t$) vs $\phi$/V of
$1/H_{eff}RT$, where R is the ideal gas constant, and T the temperature (K), and an x-intercept of $k$, the first-order loss
rate (s$^{-1}$). In practice, the linearity of this relationship and the performance of the measurement at different liquid
volumes and flow rates that result in the same $\phi$/V provide a check on the assumption of rapid equilibration within
the reactor. In practice we measure the effective Henry's coefficient in our experiments, but the distinction is only
important for the weak acid, HNCO, as described in the Results and Discussion section below.

Attempts to measure the solubility of ICN with the glass bubbler system described above were

unsuccessful, because ICN did not equilibrate at the levels and timescales typical of the other compounds measured
in this work, and the decay profiles were not reproducible nor exponential. The possibility that this was due to
higher solubility, faster reaction, or decomposition of ICN on glass surfaces was explored by using a smaller reactor
fabricated from 12.7 mm O.D. PFA tubing and PFA compression fittings (see supplemental Figure S1). In these
experiments, liquid volumes of between 1.0 and 2.0 cc and flow rates of 100 to 600 ambient cc/min were used. This
resulted in equilibration and decay profiles more similar to the other experiments, when the solubility of ICN in
water was measured at room temperature. Attempts to measure ICN solubilities in n-octanol were not successful
using either reactor.

Solution for the aqueous solubility/reaction experiments were prepared from reagent-grade materials. The

pH 2-4 buffer solutions were commercial preparations, made from citric acid monohydrate with differing amounts
of hydrochloric acid, sodium chloride, and sodium hydroxide (Fixanal, Fluka Analytical), having anion
concentrations ranging from 0.08 to approximately 0.2 M.  The manufacturer specifications (Fluka, Sigma-Aldrich)
of the pH=3 buffer showed a slight temperature dependence, with the pH ranging from 3.03 at 0 °C to 2.97 at 90 °C.
An ammonium chloride solution of 0.45 M was prepared through addition of a measured amount of the solid to the
pH=3 buffer. Sodium chloride solutions ranging up to 2.5 M were prepared gravimetrically in the pH buffer
solution. The pHs of NH$_4$Cl and NaCl solutions were measured at room temperature with a pH meter and found to
be within 0.1 pH unit of the nominal buffer pH value.

**III. Results and Discussion**

Examples of the data generated by equilibration experiments are shown in Figures 1 and 2, which show the

exponential decays for a series of gas flow rates (Figure 1) and the correlation of the decay rates versus $\phi$/V (Figure
2). Numerous other examples of both decay curves and decay rate versus $\phi$/V are shown in the Supplementary
Material for a range of different analytes and solutions. The uncertainties in the Henry's coefficients are derived
from a combination of the reproducibility of the decay rates, the agreement between decay rates at the same $\phi$/V (but
different flows and liquid volumes) and the fits to the slope of relationships like those shown in Figure 2, and were
generally ±10% or better. The uncertainties in first-order loss rate are the corresponding uncertainties in the
intercepts. The results of the experiments with HNCO, CH$_3$NCO, ClCN, BrCN, and ICN with the variety of solvents
and conditions employed are summarized in Tables 1&2 and described below.

A. Results for Aqueous Solution

1.   Solubility and Reactions of HNCO

Here we report results for pHs between 2 and 4, and for the temperature range 279.5 to 310.0 K at pH=3. In

addition, we report data for the effect of salt concentrations on the solubility at pH=3, and the effect of ammonium
concentrations on solubility and apparent first-order loss rate in solution. The dependence of aqueous solubility of
HNCO on pH is expected given it is a weak acid, and its dissolution is accompanied by an acid-base equilibrium;

HNCO$_g$ $\leftrightarrow$ HNCO$_{aq}$                H = [HNCO]$_{aq}$/[HNCO]$_g$          Eq. (2)

HNCO$_{aq}$ + H$_2$O$_{aq}$ $\leftrightarrow$ H$_3$O$^+$ + NCO$^-$       K$_a$ = [NCO$^-$][H$^+$]/[HNCO]        Eq. (3)


so that what is measured is the effective Henry's coefficient, H$_{eff}$, which involves the sum of all forms of HNCO in
solution:

H$_{eff}$ = {[HNCO]$_{aq}$ + [NCO-]}/[HNCO]$_g$                   Eq.(4)


Substituting for [NCO$^-$] using the rearranged form of Eq(3), and using Eq(2) we get the relationship for H$_{eff}$:

H$_{eff}$ = H(1 + K$_a$/[H$^+$])                             Eq. (5)


The plot of H$_{eff}$ vs 1/[H+] is shown in Figure 3, the slope of which is H×K$_a$, and the intercept is the intrinsic Henry's
Law constant, H. The resulting fit (R$^2$ = 0.99) gave a H = 20 (±2) M/atm, and a K$_a$ of 2.0 (±0.3) x10$^{-4}$ M (which
corresponds to $pK_a = 3.7 \pm 0.1$). The uncertainties in these numbers were derived from the standard deviations of the
fitted parameters, where the value for $K_a$ is the propagated uncertainty in both H and the slope. Figure 4 shows the
comparison of the H measurements from this work with those of Borduas et al., (2016) plotted according to Eq. 5
equation. There are approximately 20% differences in the two data sets, which is just at the limits of the quoted
uncertainties, when both the uncertainties in the intrinsic H and pKa are taken into account.
The temperature dependence of the solubility measured at pH=3, obeys the simple Van't Hoff relationship;

$d\ln H_{eff}/d(1/T) = - \Delta H_{soln}/R$                 Eq. (6)

shown in Figure 5 as a linear relationship of log H vs. 1/T. These data were not corrected for the slight dependence
of the buffer pH on temperature (3.02-2.99 pH units over this range).  The slope of the correlation yields a $\Delta H_{soln}$ of
-37.2 ±3 kJ/mole, calculated using dimensionless Henry's coefficients ($H_{eff}RT$), (Sander, 2015). This enthalpy of
solution agrees with that measured by Borduas et al., (2016) (-34 ±2 kJ/mole) within the stated uncertainties.
Moreover, this enthalpy is similar to those of other small N-containing molecules: HCN (-36.6 kJ/mole), $CH_3CN$
(-34.1 kJ/mole), and nitromethane (-33.3 kJ/mole) (Sander, 2015), but different than that of formic acid (HC(O)OH)
(-47.4 kJ/mole) which was used by the cloud uptake modeling study (Barth et al., 2013).
Often the Henry's Law solubility can depend on salt concentration of the solution, usually resulting in a
lower solubility (salting out), but occasionally resulting in a higher solubility (salting in), with higher salt
concentrations. These effects are most applicable to aerosol chemistry, where ionic strengths can be quite high. This
effect on HNCO solubility was measured at pH=3 and 298 K for NaCl solutions between 0 and 2.5 M concentration.
The results, shown in Figure 6, exhibit the classic "salting out" effect where HNCO was only about 60% as soluble
at 2.5 M compared to the standard pH=3 buffer. The Setschenow constant, $k_s$, can be determined by the relationship:

$-\text{Log}(H_{eff}/H_{eff0}) = k_s \text{ x}[I]$               Eq. (7)

where $H_{eff}$ is the Henry's coefficient at a given ionic strength, $I$, and $H_{eff0}$ is the Henry's coefficient in pure water.
For a salt with two singly charged ions, $I$ is equal to the salt concentration. In this experiment, $k_s$ was found to be
$0.097 \pm 0.011$ $M^{-1}$. The magnitude of the salting out effect on HNCO is similar or slightly smaller than those found
for other small organic compounds in NaCl, such acetylene, ethane and butane (Clever, 1983; Schumpe, 1993),
Interestingly, (Wang et al., 2014) found that Setschenow constants for ammonium sulfate {$(NH_4)_2SO_4$} are typically
larger than those for NaCl, a feature which might impact the uptake of HNCO to aerosol particles having substantial
$(NH_4)_2SO_4$ content.
The net hydrolysis reaction rates observed in this study are listed in Table 1, and range from 0.22 to 4.15
$\times 10^{-3}$ $s^{-1}$ and are both pH and temperature dependent. The main reactions of HNCO/NCO$^-$ in aqueous solution are
hydrolysis reactions that involve the acid or its conjugate anion, as detailed in Reactions 1-3 noted above. The
expression for the net hydrolysis reaction is;

$$k_{hydr} = \frac{k_1[H^+]^2 + k_2[H^+] + k_3 K_a}{K_a + [H^+]}$$        Eq. (8)

as derived by Borduas et al., 2016. The rates of these reactions that were determined in several previous studies
(Borduas et al., 2016; Jensen, 1958) and are in reasonable agreement except for Reaction 3, which is not
atmospherically relevant. Equation 8 was used to calculate the values from those two studies that would correspond
to the rates at pH=3 measured in our work, and are also listed in Table 1. The rate constants reported in this work
agree within the range observed in the two previous studies, except for one temperature, and the relative standard
deviations of mean values calculated from all three observations ranged from 5 to 30%.

The above hydrolysis reactions represent a lower limit on the condensed phase loss of HNCO, so reaction

with other species present in the condensed phase might result in faster loss, and produce unique chemical species.
HNCO/NCO$^-$ are known to react with a variety of organic compounds having an "active hydrogen" (a hydrogen
attached to an O, N, or S atom)(Belson and Strachan, 1982), through simple addition across the N=C bond, where
the active hydrogen ends up on the N and the other moiety ends up attached to the carbon. For example, alcohols
react to yield carbamates, i.e. esters of carbamic acid:

HNCO + ROH → H$_2$NC(O)OR        (R10)


Note that this is really the same mechanism as the neutral hydrolysis of HNCO, except that the addition of water
forms carbamic acid, H$_2$NC(O)OH which is unstable and decomposes to NH$_3$ and CO$_2$. In the same fashion,
HNCO/NCO$^-$ can react with ammonia in solution to yield urea

HNCO + NH$_3$ ↔ H$_2$NC(O)NH$_2$        (R11)


And in a more general sense, react with amines to yield substituted ureas:

HNCO + RNH$_2$ ↔ H$_2$NC(O)NHR        (R12)


Reaction 11 is known to be an equilibrium that lies far to the product side under all conditions pertinent to this work
(Hagel et al., 1971). While the forward reaction rate for R11 has been measured under neutral to slightly basic
conditions (Jensen, 1959; Williams and Jencks, 1974b), it has not been measured at pHs applicable to atmospheric
aerosol or cloud droplets, i.e. pH=2-4. These previous studies have assumed that the mechanism involves the
reaction of the un-ionized species, e.g. NH$_3$ and HNCO, although there is some evidence that Reaction 12 for some
amines (RNH$_2$) has a more complicated reaction mechanism (Williams and Jencks, 1974a). As a consequence of this
assumption, the previous studies reported their reaction rates corrected for the acid-base equilibria of each species.
The solubility/reaction experiment in this work was performed at pH=3 and [NH$_4^+$] of 0.45 M, so a substantial
correction of the literature values for the acid-base equilibria in the case of NH$_4^+$ and a minor correction for the
dissociation of HNCO was required in order to compare with our result. The results of our study (Table 1) show that
the solubility of HNCO in $NH_4Cl$ solution at 292 K is essentially the same as that of the pH=3 buffer alone (31.5 ±3
vs 32.6 ±3 M/atm). This implies that R11 does not impact the aqueous solubility. However, the measured first-order
loss rate, 1.2 (±0.03) $\times10^{-3}$ $s^{-1}$ is faster than the hydrolysis at pH3, 0.66 (±0.06) $\times10^{-3}$ $s^{-1}$. The reaction can be
expressed as the sum of hydrolysis and reactions of HNCO and $NCO^-$ with $NH_4^+$ (the predominant form at pH3).
$\frac{d[HNCO+NCO]}{[HNCO+NCO]} = -(k_{hydr} + k_{11}[NH_4^+])dt$                          Eq. (9)
We calculate a value of 1.2 (±0.1) $\times10^{-3}$ $M^{-1}s^{-1}$ for $k_{11}$ from our measurements which is much faster than the rate
constants reported by previous studies, $5 \times10^{-6}$ $M^{-1}s^{-1}$ (Jensen, 1959) and $1.5 \times10^{-5}$ $M^{-1}s^{-1}$ (Williams and Jencks,
1974b), when corrected for acid-base equilibria.

2.    Solubility and Reactions of $CH_3NCO$


The solubility and first-order loss rate of $CH_3NCO$ were measured at pH=2 and pH=7 at 298 K, and the

results are listed in Table 1. The Henry's coefficients, 1.3 (±0.13) and 1.4 (±0.14) M/atm, were lower than those
measured for HNCO, and independent of pH, within the uncertainties of the measurements. This is consistent with
MIC being a less polar compound, with no dissociation reactions that might be pH dependent. In addition, these
results imply that solution complexation due to the presence of anions does not affect MIC solubility, at least that
the concentrations and anions present in the pH=2 buffer solution, 0.2 M for the sum of citrate and chloride.

The first-order loss rates of MIC, presumably due to hydrolysis, did show a pH dependence that implies

acid catalysis. These hydrolysis rates were faster than the rates for HNCO at the same temperatures and pHs. The
mechanism of $CH_3NCO$ hydrolysis and other solution chemistry is discussed by (Al-Rawi and Williams, 1977;
Castro et al., 1985). The hydrolysis of $CH_3NCO$ is thought to proceed first by formation of a methyl carbamic acid:

$CH_3NCO + H_2O \leftrightarrow CH_3NHC(O)OH$                          (R13)


which is analogous to the way water adds across the N=C bond of HNCO. The methyl carbamic acid then either
reacts with $H_3O^+$ (faster) or $H_2O$ (slower) to produce $CH_3NH_3^+$ and $CO_2$, or $CH_3NH_2$ and $CO_2$, yielding the net
reactions (R4) and (R5):

$CH_3NCO + H_3O^+ \rightarrow CH_3NH_3^+ + CO_2$                         (R4)

$CH_3NCO + H_2O \rightarrow CH_3NH_2 + CO_2$                           (R5)


The Henry's law measurements in our work imply that if (R13) is happening, it must be to a quite minor extent,
otherwise the H constant for $CH_3NCO$ would be much larger than it is. Solution-based studies of MIC in the
presence of strong acid anions (Al-Rawi and Williams, 1977; Castro et al., 1985) also imply that a complex
mechanism takes place, involving a reversible complexation, (shown here for $HSO_4^-$):

$CH_3NCO + HSO_4^- \leftrightarrow CH_3NH\text{-}C(O)\text{-}OSO_3^-$                     (R14)


Rate constants for reactions R4 and R5 were reported by Castro et al., (1985) but the precision of these were
somewhat compromised by the presence of the R14 equilibrium. Again, in this study, the Henry's coefficient results
imply a negligible role for complexation, so the following simplified expression for the pH dependence is used:

$$k_{MIC} = k_5 + k_4[H_3O^+] \hspace{4cm} (Eq\ 10)$$

to derive the following values for $k_5 = 1.9\ (\pm0.6) \times10^{-3}\ s^{-1}$, and $k_4 = 0.13\ (\pm0.07)\ M^{-1}s^{-1}$. These values are in
reasonable agreement with the value for $k_5$ given by Al-Rawi and Williams, (1977), $1.47\ x10^{-3}\ s^{-1}$ considering those
measurements were at 1M KCl, and the value for $k_4 = 0.16\ M^{-1}s^{-1}$ given by Castro et al., (1985) for reaction with
HCl in the absence of a buffer.

3.    Solubility and Reactions of XCN Compounds.


The solubilities and first-order loss rates of XCN compounds were measured at room temperature and

neutral pH in pure DI water, and at ice/water temperature for ClCN and BrCN. The resulting Henry's coefficients
are listed in Table 2. The ClCN solubility was essentially the same as that measured for MIC at room temperature,
and is in reasonable agreement with the value of 0.52 M/atm at 298 K based on a model estimate (Hilal et al., 2008),
and one reported measurement, 0.6 M/atm at 293 K (Weng et al., 2011).  In contrast, BrCN was more soluble than
ClCN, $8.2 \pm0.8$ M/atm at 296°K, but fairly insoluble in an absolute sense. The temperature dependences of $H_{ClCN}$
and $H_{BrCN}$ were as expected, showing higher solubility at lower temperatures, however, they had very different heats
of solution, -27.8 kJ/mole for ClCN, and -38.3 kJ/ mole for BrCN, although there are only two data points for each
compound. Both the higher solubilities and larger $\Delta H_{soln}$, could be a result of the higher dipole moment and
polarizability of BrCN relative to ClCN (Maroulis and Pouchan, 1997).

The solubility of ICN was measured a room temperature using a combination of different flow rates (208 –

760 amb cm$^3$/min) and liquid volumes (1.95 and 0.95cc). A plot of the decay rates versus $\phi$/V for those runs is
shown in Figure 7. Where those data sets overlap there is agreement to within about 15%, implying that the
equilibration could be fast enough to meet the criteria for these types of flow experiments. The resulting Henry's
coefficient, 270 ($\pm$41) M/atm is significantly larger than the other two XCN compounds, but is consist with the trend
of increasing solubility with dipole moment and polarizability. Attempts to use the small reactor to measure the
solubility of ICN at ice/water temperatures was not successful, e.g. did not yield simple single exponential decays
with time under the same range of flow conditions as used in the room temperature experiment.

The hydrolysis of XCN compounds is known to be base-catalyzed, and can be susceptible to anion

complexation (Bailey and Bishop, 1973; Gerritsen et al., 1993) in a manner similar to MIC:

$XCN + A^- \leftrightarrow [XCN{\bullet}A]^- \hspace{5cm} (R15)$

$XCN + H_2O \rightarrow HOCN + H^+ + X^- \hspace{4cm} (R6)$

$XCN + OH^- \rightarrow HOCN + X^-$                                                    (R7)


This complexation can be ignored in our study for ClCN and BrCN since the experiment was performed in

DI water, however, such complexation should be considered in future condensed phase studies of XCN compounds.
Accordingly, the expression for the ClCN and BrCN hydrolysis rate constant is;

$k_{XCN} = k_w + k_{OH}[OH^-]$                                                        (Eq.11)


Bailey and Bishop (1973) found $k_w = 2.58 \times 10^{-6}$ s$^{-1}$ and $k_{OH} = 4.53$ M$^{-1}$s$^{-1}$ at 299.7 K, for ClCN, which corresponds
to $3.03 \times 10^{-6}$ at pH7. This is consistent with the results of this study which found that the first-order loss rate was
zero, within the error of the linear fit ($\pm 4.2 \times 10^{-4}$ s$^{-1}$). The study of BrCN hydrolysis of Gerritsen et al., 1993 did not
derive $k_w$ nor did it present sufficient data for $k_w$ to be estimated. However, there are two other studies that presented
data from which $k_w$ can be estimated, and those range from $1.9 – 9.2 \times 10^{-5}$ s$^{-1}$, (Heller-Grossman et al., 1999;
Vanelslander et al., 2012).

The hydrolysis of ICN is slightly more complicated since there is some evidence that ICN might complex

with iodide (Gerritsen et al., 1993). The room temperature hydrolysis rate observed in our experiment was not
significantly different than zero, $4.4 (\pm 7.6) \times 10^{-5}$ s$^{-1}$, but is in the same range of the rate constant estimated from the
data given by Gerritsen, et al., (1993), by extrapolating their rate constant vs. [OH$^-$] data to zero [OH$^-$], assuming no
complexation reactions.

B. Non-aqueous Solution

Solubility in non-aqueous solvents is a standard indicator of how compounds will be distributed between

different compartments in the environment, i.e. lipids in the body, organic aerosols in the atmosphere. In addition,
the ratio of organic to aqueous solubility ($K_{ow}$) is used to estimate membrane transport of a chemical species, a key
factor in estimating physiologic effects of a pollutant. Several non-aqueous solvents were used in this study,
tridecane to represent a completely non-polar solvent and n-octanol, which is used as a standard material for such
studies. Tridecane was used because it is the heaviest n-alkane that is still a liquid at 273.15K, and it has purely non-
polar character, i.e. no functional groups, so is a slightly different model for non-polar matrices.

1. Solubility and Reactions of HNCO


The experiments performed on HNCO were conducted with tridecane, 10% (V/V) n-octanol/tridecane, and

pure n-octanol, and the results are summarized in Table 1. HNCO is the least soluble in tridecane, 1.7 ($\pm 0.17$) M/atm
and increasingly soluble as the proportion of n-octanol is increased, to pure n-octanol, 87 ($\pm 9$) M/atm at 298 K.
Experiments at two other temperatures were performed to confirm that these solubilities follow the expected
temperature dependence, and to obtain the solubility in pure n-octanol at human body temperature (310 K) to match
data for the aqueous solubility. The lower solubility of HNCO in tridecane is expected since tridecane is completely
non-polar and has no tendency to hydrogen bond or interact with the polarizable end of the HNCO molecule. In
contrast, the increase in solubility of HNCO with increasing proportion of n-octanol is due to the polar –OH group at
the end of the molecule.
The rate of reactions of HNCO with the non-aqueous solvents were below the limit of detection by this
method for all combinations except for pure n-octanol at 310 K. Even still, the measured rate was quite a bit lower
than the corresponding hydrolysis rate in aqueous solution at pH=3. The manner in which these two factors
(solubility and reaction) affect the net uptake and loss of HNCO will be discussed below.

2. Solubility and Reactions of $CH_3NCO$

The solubility of $CH_3NCO$ in n-octanol was measured at several temperatures, as summarized in Table 1. The
value for 298 K is approximately 3 times higher than that of aqueous solubility, and has the expected temperature
dependence. In addition, the first-order reaction rates for $CH_3NCO$ in n-octanol were in the same range or slightly
higher than the aqueous reactions. The reaction with n-octanol is expected to go via the carbamylation reaction
(R10), although there is some evidence that this reaction has as more complex mechanism possibly involving
multiple alcohol molecules (Raspoet et al., 1998). These rates are much faster than the corresponding rates for
HNCO, and may provide some guidance concerning the loss rates of $CH_3NCO$ to heterogeneous processes.

3. ClCN and BrCN

The solubilities of ClCN and BrCN in n-octanol were measured at room temperature. Cyanogen chloride and
BrCN have about the same relative differences in solubility in n-octanol (a factor of 3-4) as they did $H_2O$. The higher
solubility of BrCN relative to ClCN could again be due to its higher dipole moment and polarizability (Maroulis and
Pouchan, 1997).  The first-order loss rates of ClCN and BrCN could be determined from the flow reactor experiments
and were 1.3 ($\pm0.4$) $\times10^{-3}$ s$^{-1}$ and 9 ($\pm2$) $\times10^{-5}$ s$^{-1}$, respectively. Reactions of ClCN with alcohols are known (see for
example (Fuks and Hartemink, 1973)), and form carbamates, in a mechanism that appears to be second-order in the
alcohol, and acid catalyzed, but rate constants for ClCN-alcohol reactions have not been reported to our knowledge.
There are studies of rates of reactions of ClCN with nucleophiles, e.g. nitrogen bases, and those reactions appear to
result in –CN substitution and formation of a Cl$^-$ ion (Edwards et al., 1986). In addition, BrCN has been used by
protein chemists to selectively cleave disulfide bonds and has been used for some time by synthetic chemists to
selectively convert tertiary amines to secondary amines (Siddiqui and Siddiqui, 1980; von Braun and Schwarz, 1902)
and can carbamylate amino acids (Schreiber and Witkop, 1964). The importance of these reactions to the atmospheric
fate of XCN compounds remains an open question, but it is important to note that they constitute losses of active
halogen, i.e. conversion of the halogen to a halide ion.

4. Octanol/Water Partition Coefficients

The ratio of solubilities between a non-polar solvent and water is a fundamental quantity that is useful in

predicting the fate of a compound in the environment and biological systems (Leo et al., 1971). This parameter is
used to predict lipid solubility, membrane transport, and the potential of uptake of a particular compound by organic
aerosol. n-Octanol is a standard non-polar solvent that is commonly used for this purpose, as it has an overall non-
polar character with a substituent that is capable of hydrogen bonding. The data from this study can be used to
calculate the octanol/water partition coefficients for HNCO, $CH_3NCO$, ClCN, and BrCN as the ratio of the
respective Henry's coefficients;

$K_{ow} = H_{oct}/H_{H2O}$                                                            (Eq.12)

The results are listed in Table 4 along with $K_{ow}$s for some related small molecules. Both $CH_3NCO$, and BrCN are
fundamentally more soluble in n-octanol than in water, while ClCN has nearly the same solubility in both materials.
The weak acid equilibrium of HNCO makes it more soluble in n-octanol at pH 3, but much more soluble in water at
neutral pH. However, transport models of biological systems account for these acid base equilibria along with using
the $K_{ow}$ to estimate transport rates (Missner and Pohl, 2009). Formic acid is a similarly weak acid (pKa = 3.77) and
so is a good point of comparison to HNCO. The n-octanol partition coefficient of HNCO is a factor of 15 larger than
that of HC(O)OH, so should have larger membrane permeabilities. Similarly, the n-octanol partition coefficient of
$CH_3NCO$ is 6.8 times larger than that of $CH_3CN$. The two cyanogen halides measured here had differing behavior,
with ClCN showing almost no difference in solubility, and BrCN having about the same increase in solubility in n-
octanol as HNCO and $CH_3NCO$.
**IV. Atmospheric and Environmental Chemistry Implications**

The atmospheric loss of the compounds studied here are either solely or predominantly through

heterogeneous uptake and reaction for HNCO, $CH_3NCO$, ClCN, BrCN, or in the case of ICN due to both
heterogeneous chemistry and photolysis. The aqueous solubility and reaction data from this study allow some
prediction of uptake parameters and loss rates in some important systems, e.g. cloud water and natural water
surfaces like oceans. In addition, some indications can be gained about the uptake of HNCO, $CH_3NCO$, ClCN, and
BrCN to organic aerosol, using n-octanol as a model. Finally, the n-octanol/water partition coefficient is often used
as a key parameter in modeling cross-membrane transport, and the data from this study can be used to predict the
behavior of these reduced-N compounds relative to other well-studied compounds.

The reactive uptake of HNCO, $CH_3NCO$ and XCN on environmental surfaces, small particles and aqueous

droplets can be parameterized using the uptake coefficient, $\gamma$, defined as the fraction of collisions of a molecule with
a surface that lead to incorporation of that molecule in the condensed phase. If solubility and reaction are the
limiting processes, a good assumption for the species in this work, then $\gamma_{rxn}$ can be estimated from the following
equation (Kolb et al., 1995):

$\gamma_{rxn} = \frac{4HRT\sqrt{kD_a}}{<c>}$                                                        (Eq.13)

where H and k are the Henry's coefficient and first-order loss rate in solution measured in this work, R is the gas
constant, T is temperature, $D_a$ is the diffusion coefficient in aqueous solution (assumed here to be $1.9 \times 10^{-5}$ cm$^2$ s$^{-1}$
for HNCO and $1.6 \times 10^{-5}$ cm$^2$ s$^{-1}$ for CH$_3$NCO and ClCN, $1.2 \times 10^{-5}$ cm$^2$ s$^{-1}$ for BrCN, and $1.0 \times 10^{-5}$ cm$^2$ s$^{-1}$ for ICN at
298 K), and <c> is the mean molecular velocity. The results of these calculations are shown in Figure 8 for the H
measurements at 298K reported here, $k_{hydr}$ for HNCO reported by Borduas et al., (2016), $k_{hdyr}$ for CH$_3$NCO from this
work and $k_{hydr}$ for ClCN from Bailey and Bishop (1973).

Deposition of a compound to the surface can be parameterized as essentially two processes taking place in

series, physical transport within the planetary boundary layer to the surface and then chemical uptake on the surface
(see for example (Cano-Ruiz et al., 1993)). In this formulation, the deposition velocity, $v_d$ (the inverse of the total
resistance) is expressed as follows:

$$v_d = \frac{1}{\frac{1}{v_t} + \frac{1}{\gamma \frac{<c>}{4}}}$$
(Eq.14)


where $1/v_t$ is the resistance due to transport, and $\frac{1}{\gamma \frac{<c>}{4}}$ is the resistance due to chemical uptake. For a species for
which uptake is rapid, e.g. a highly soluble acid, the chemical resistance becomes small and $v_d \cong v_t$. This is the case
for HNCO deposition to land or natural water surfaces (pHs ~7-8). Typical $v_t$s are on the order of 0.5 to 1 cm/s for a
reasonably mixed boundary layer (Wesely and Hicks, 2000). For compounds for which $\gamma$ is quite small, the
chemical term predominates.
$v_d \cong \gamma \frac{<c>}{4}$                                        (Eq.15)
The lifetime of a species within the PBL then can be estimated as $h/v_d$, where h is the boundary layer height. The
lifetime estimates for HNCO, CH$_3$NCO, and XCN compounds are given in Table 3, and range from the short
lifetime noted for HNCO, to quite long lifetimes for the least soluble species, for example ClCN.

The loss rates due to uptake of species to atmospheric aerosol particles can be estimated from the pH

dependent uptake coefficients in Figure 8, using parameterizations described in the literature (Davidovits et al.,
2006; Sander, 1999). In the limited case of surface-controlled uptake, i.e. neglecting gas phase diffusion, the loss of
a species is;

$k = \frac{A\gamma <c>}{4}$                                           (Eq.16)

where A is the aerosol surface area. If we take the $\gamma$s from Figure 8, and assume highly polluted conditions to obtain
a lower limit to the lifetime against this process, A = 1000 μm$^2$/cm$^3$ and pHs between 1 and 2, then the lifetimes
listed in Table 3 are arrived at. The values for HNCO and CH$_3$NCO show a range because the uptake is pH
dependent, and it should be noted that the values for CH$_3$NCO, ClCN, and BrCN are over estimated by this method,
as their chemistry is slow enough that a volume-based estimate may be more appropriate. The more important effect
here is that the $\gamma$ values are based on hydrolysis losses, which are undoubted much slower than many of the solution-
phase reactions that these species can undergo, hence the lifetimes against aerosol deposition are upper limits.
The loss of HNCO to cloudwater is the subject of extensive work discussed by Barth et al, (2013), and no
attempt will be made here to update that analysis. We can point out that our results yielded slightly lower $H_{eff}$
(~22%) at the lowest temperature we measured, compared to the values used by Barth et al., (2013), see Figure S13.
This would result in slightly slower removal rates in the Barth et al., model in low-temperature clouds. The fastest
loss rates for HNCO were observed in warm dense clouds into which $SO_2$ was also dissolving and adding
considerable acidity, so that value for HNCO was included in Table 3. For the other compounds we use a simple
parameterization of cloudwater reaction to estimate the in-cloud loss rates for $CH_3NCO$ and the XCN compounds. In
the estimate of reaction rate:
$\quad\quad k = k_l L_c HRT$ (Eq.17)
$k_l$ is the liquid phase rate constant, $L_c$ is the cloud liquid water content, and H is the Henry's coefficient. If we
assume a $L_c$ of 2 x$10^{-6}$, and T $\cong$298 K, and we use the H and k values measured in this work (the exception is that
the literature value for $CH_3NCO$ at pH=2 was used), then the values for lifetimes of $CH_3NCO$, and XCN compounds
listed in Table 3 were obtained.  Below we discuss the characteristic times obtained for each compound in the
context of what else is known about their sources and atmospheric chemistry.
A. HNCO
The loss of HNCO via heterogeneous processes occurs in two separate regimes: in aerosols and cloud
droplets at relatively low pH, and in surface waters and on terrestrial surfaces that are neutral or slightly basic in pH.
In the former case, HNCO solubility is relatively low but hydrolysis is acid catalyzed. In the latter case, solubility is
high enough that uptake will be limited by the transport of HNCO to the surface, much like other strong acids such
as $HNO_3$. Ambient measurements of HNCO at surface sites are consistent with deposition of HNCO to the ground,
exhibiting diurnal profiles similar to those of $O_3$ or $HNO_3$ (Kumar et al., 2018; Roberts et al., 2014; Mattila, et al.,
2018; Zhao et al., 2014).
Several aspects of the aqueous solubility and hydrolysis, and heterogeneous removal of HNCO have been
examined in modeling studies.  A global modeling study by Young et al. (2012) was a first attempt to model global
HNCO by scaling the source to fire emissions of HCN. Loss of HNCO was assumed to be due to wet and dry
deposition with efficiencies similar to $HNO_3$ and $HC(O)OH$, and that HNCO was lost once it was taken up by
clouds. Young et al., concluded that HNCO had an average lifetime of about 37 days.  Barth et al., (2013) addressed
part of this analysis by modeling the cloud removal of HNCO using actual solubility and reaction data in a cloud
parcel model, albeit, the hydrolysis rates used were from Jensen, (1957) which were approximately 50% higher than
the Borduas results, and the temperature dependence of $H_{eff}$ was assumed equal to that of $HC(O)OH$, and resulted in
higher solubilities at low temperature. This cloud model showed that cloud water uptake was reversible in that most
cases hydrolysis was slow enough that some HNCO returned to the gas phase after cloud evaporation. The Barth et
al., study estimated HNCO lifetimes as short as 1 hour in warm polluted clouds (i.e. high $SO_2$ => $H_2SO_4$ formation).
The results of our study and those of Borduas et al., (2016) add to these analyses in that now the measured
temperature dependence of $H_{eff}$ can be used, and the hydrolysis rate constants can be updated.
The results in this paper allow for further refinement of HNCO loss estimates. For example, the salting-out
effect may be important for aerosol with high inorganic content, and high ammonium concentrations will result in
reactive loss rates that are faster than hydrolysis. The solubility of HNCO in aerosol particles with substantial
organic character can be higher or low depending on the nature of substituent groups, e.g. degree of -OH
functionalization. Given that aerosol particles in most polluted atmospheres are at least half organic carbon by mass
(Jimenez and et al., 2009), it is useful to estimate what effect an increased solubility of HNCO might have on its
removal lifetime. If take our 1000 $\mu m^2/cm^3$ surface area aerosol from the above calculation, assume a 50/50 organic
to aqueous distribution and that the solubility of HNCO in the organic fraction is the same as n-octanol, we can
arrive at a weighted average Henry's solubility of 55 M/atm. If we combine that with the same reaction rate
corresponding to pH1, then the lifetime of HNCO against reaction to this aerosol drops to about 2 days, a significant
effect.
In studies of the condensed phase oxidation of dissolved N species, as well as biological processes produce
cyanate ion, there is a growing recognition that cyanate is part of the natural N cycle in the ocean (see (Widner et al.,
2013) and references there-in). Observed near-surface cyanate levels often reached a few 10s of nM in near shore
productive areas.  The observations of cloud/aerosol source of HNCO presented in (Zhao et al., 2014) on the coast
of California might be explained by a combination of this $NCO^-$ seawater source and aerosol/cloud water
acidification by local sources of strong acids, particularly $HNO_3$. In specific, acidification of sea spray containing
about 10 nM $NCO^-$ to pH=4 or so, would correspond to $H_{eff}$ of around 50 M/atm, and result in an equilibrium
HNCO concentration of several hundred pptv. Such a source would most likely be limited by the concentration of
sea salt-derived aerosol, but could easily account for the source implied by the measurements of (Zhao et al., 2014).
B. $CH_3NCO$
The atmospheric chemistry of $CH_3NCO$ is less well studied than HNCO. There is a single reported
measurement of the reaction rate of $CH_3NCO$ with OH by relative rates which gave $k = 3.6 \times 10^{-12}$ $cm^3$ $molec^{-1}$ $s^{-1}$
(Lu et al., 2014), however recent work indicates that secondary chemistry may have made this rate high by a
significant amount (Papanastasiou, et al., manuscript in preparation, 2019).  In addition, there are likely condensed-
phase reactions that are faster than the simple hydrolysis reactions consider in this work. Never-the-less, it is useful
to estimate the atmospheric loss rates implied by our work, as a baseline against which future atmosphere
observations can be judged, and the importance of other heterogeneous processes can be assessed.  The uptake
coefficients estimated for $CH_3NCO$ in Figure 8 are relatively low with only a slight increase at the lowest pHs in
atmospheric media. As a consequence, atmospheric lifetimes of $CH_3NCO$ towards surface deposition are estimated
to be quite long, 6 months or more if hydrolysis is the sole loss process. The loss due to aerosol of cloudwater
uptake is estimated to be slightly faster, due primarily to the slight acid-catalysis of the $CH_3NCO$ hydrolysis rate.
C. ClCN, BrCN, and ICN
To date, we know of no observations of ClCN in the ambient atmosphere, but its formation in the
chlorination of water, waste water, and swimming pools (Afifi and Blatchley III, 2015; Daiber et al., 2016; Lee et
al., 2006) indicates that there could be sources from human activities, including the use of chlorine bleach for
cleaning indoor surfaces.  In addition, there might also be a source from aerosol systems were chlorine is being
activated, i.e. oxidized from $Cl^-$ to $ClNO_2$, $Cl_2$, or $HOCl$ (see for example (Roberts et al., 2008)) in the presence of
reduced nitrogen. The results of our solubility measurements indicate that ClCN will volatilize from the condensed
phase fairly readily, e.g. within seconds of the application of a thin film of chlorine bleach cleaning solution, or the
bubbling of air though a spa in which ClCN is dissolved.  As a result, the atmospheric removal of ClCN should be
considered. BrCN has been observed in systems where bromide-containing water or wastewater were treated with
halogens (Heller-Grossman et al., 1999), and there are biological mechanisms that make BrCN and ICN as well
(Schlorke et al., 2016; Vanelslander et al., 2012). The potential for remote atmospheric sources of these compounds
is currently being investigated, but BrCN could be the result of the same bromine activation chemistry that depletes
ground level ozone in that environment (Simpson et al., 2007).

Gas phase radical reactions of XCN compounds have not been studied under atmospheric conditions. A few

studies at higher temperatures and the studies of HCN and $CH_3CN$ can be used to roughly predict how fast the
relevant reactions are. For example, the reactions of ClCN and BrCN with O atoms at 518-635 K are very slow (<3
$\times 10^{-15}$ $cm^3$ $molec^{-1}$ $s^{-1}$,(Davies and Thrush, 1968)) and the reaction of Cl atom with ClCN at high temperature is also
quite slow (<1.0 $\times 10^{-14}$ $cm^3$ $molec^{-1}$ $s^{-1}$, (Schofield et al., 1965)). However, these observations do not preclude the
presence of another reaction channel at low temperature, e.g. a mechanism involving addition to the CN group. The
reactions of HCN and $CH_3CN$ with OH, Cl atom and O atom at atmospherically relevant temperatures are all quite
slow, implying such addition channels are not likely to be substantially faster for these XCN compounds. We
conclude that rate constants for the reactions of OH or Cl with X-CN compounds are likely quite low (<2 $\times 10^{-14}$ $cm^3$
$molec^{-1}$ $s^{-1}$), making the lifetimes of these compounds against these reactions on the order of a year or longer. The
UV-visible absorption spectra of all three of these compounds have been measured (Barts and Halpern, 1989; Felps
et al., 1991; Hess and Leone, 1987; Russell et al., 1987), have maxima that range from <200nm for ClCN, 202nm
for BrCN, and 250nm for ICN, with absorption that tails into near-UV and visible wavelengths, (see Figure S13 in
the Supplemental Material). Extrapolation of the spectra, combined with photo fluxes estimated from the NCAR
TUV model for mid-summer 40° North at the surface, result in a range of photolysis behavior ranging from no
tropospheric photolysis of ClCN, to slight photolysis of BrCN ($\tau \cong 135$ days), and faster photolysis of ICN ($\tau \cong 9$
hours). The above gas phase processes provide the context in which to assess the importance of condensed phase
loss processes of ClCN, BrCN, and ICN. Rates of loss of XCN compounds due to surface deposition, cloudwater or
aerosol uptake would need to be faster than the gas phase processes to be important in the atmosphere. In addition,
condensed phase reactions convert XCN to halide ions either by hydrolysis to cyanate, or creation of a carbamyl
functionalities. Only photolysis reforms the halogen atom, and therefore maintains active halogen reaction chain.
Estimated atmospheric lifetimes of XCN compounds against loss due to condensed phase reactions listed in Table 3
shows a general trend. The lifetimes become shorter as the halogen atom goes from Cl to Br to ICN, primarily due to
higher solubilities. The actual condensed phase losses are likely much shorter than those estimated here because of
faster condensed phase reactions that are not taken into account by the brief analysis presented here. Depending on
the mechanism of condensed phase XCN reactions, this chemistry could be a condensed phase source of $NCO^-$ and
therefore HNCO similar that observed by Zhao et al. (2014) in coastal clouds.

D. Solubility in non-polar media, uptake to organic aerosol, and membrane transport.

The solubilities of HNCO, $CH_3NCO$, and BrCN in n-octanol were roughly a factor 4 larger than water, while that of ClCN was virtually the same. Reaction rates with n-octanol were the same or slower than for aqueous solutions, except for ClCN which was faster than hydrolysis at pH=7. As a result, loss due to uptake to organic aerosol will be only slightly faster for all of these species. Membrane transport is a key process in determining the extent to which a chemical species will impact biological systems. Simple membrane transport models parameterize this process as diffusion through a lipid bi-layer according to a partition coefficient, $K_p$, which the ratio of solubilities in lipid versus aqueous media (Missner and Pohl, 2009), and $K_{ow}$ is often used for this partition coefficient. The results of our work indicate that both HNCO and $CH_3NCO$ are more soluble in n-octanol than water, in contrast to other similar small organic acids and N–containing compounds (Table 4). These features will need to be accounted for in assessing the connection by between environmental exposure to HNCO, $CH_3NCO$, ClCN and BrCN and resulting biochemical effects.

**V. Data availability.** The data are available on request.

**VI. Author contributions.** YL and JR performed the laboratory experiments and JR and YL wrote the paper.

**VII. Competing interests.** The author declare no competing interests.

**VIII. Disclaimer.** Any mention of commercial products or brands were solely for identifying purposes and should not be construed as an endorsement.

**IX. Acknowledgements**

We thank Drs. James B. Burkholder, Françios Bernard, for help with the FTIR analysis and Drs. Patrick Veres and Bin Yuan for the CIMS analyses of XCN samples. We thank Dr. Dimitris Papanastasiou, Dr. J. Andrew Neuman, and Dr. Patrick Veres for communicating the results of their work prior to publication. This work was supported in part by NOAA's Climate and Health of the Atmosphere Initiatives.

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

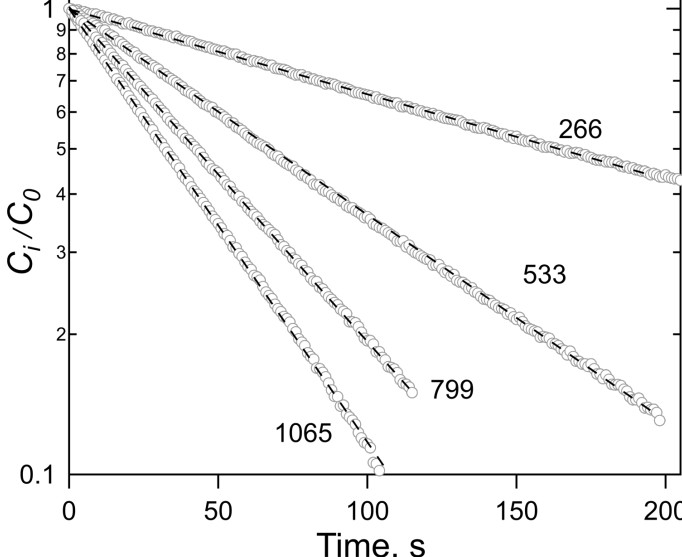

**Figure 1.** Plots of the ratio of HNCO concentration at time t, $C_i$, to the initial concentration, $C_o$, versus time for a
series of flow rates, noted as ambient cc/min. The solvent was tridecane ($C_{13}H_{28}$) and 299 K.

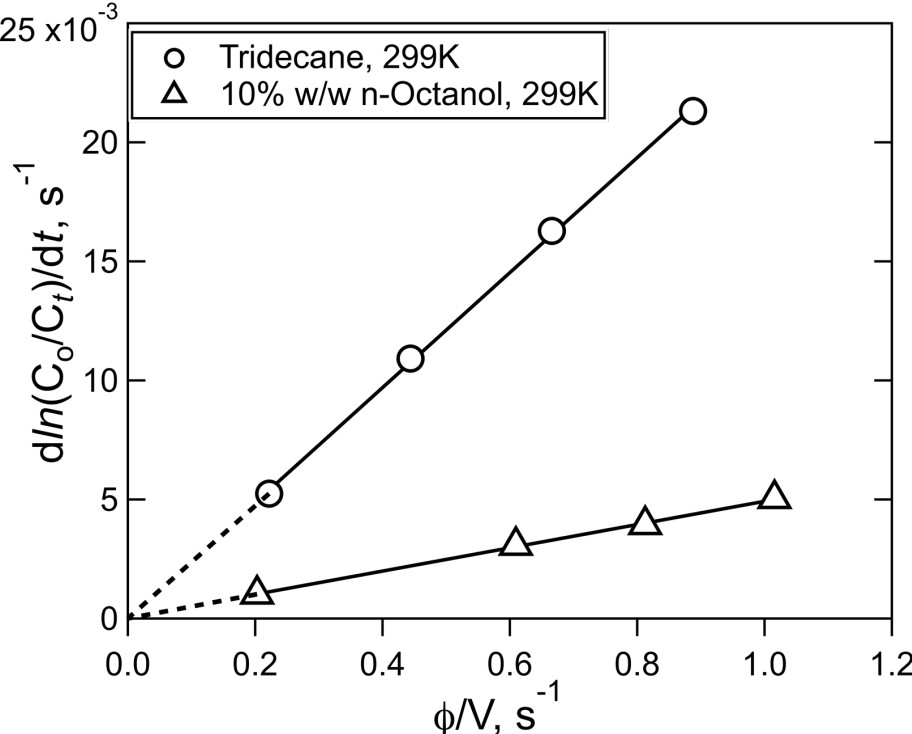

Figure 2. Plots of HNCO loss rate versus the ratio of volumetric flow rate, $\phi$, to solution volume, V, for the
experiment shown in Figure 1, (circles), and the experiment with 10% w/w n-octanol in tridecane at 299 K
(triangles). The error bars in the individual rates were smaller than the width of the points.

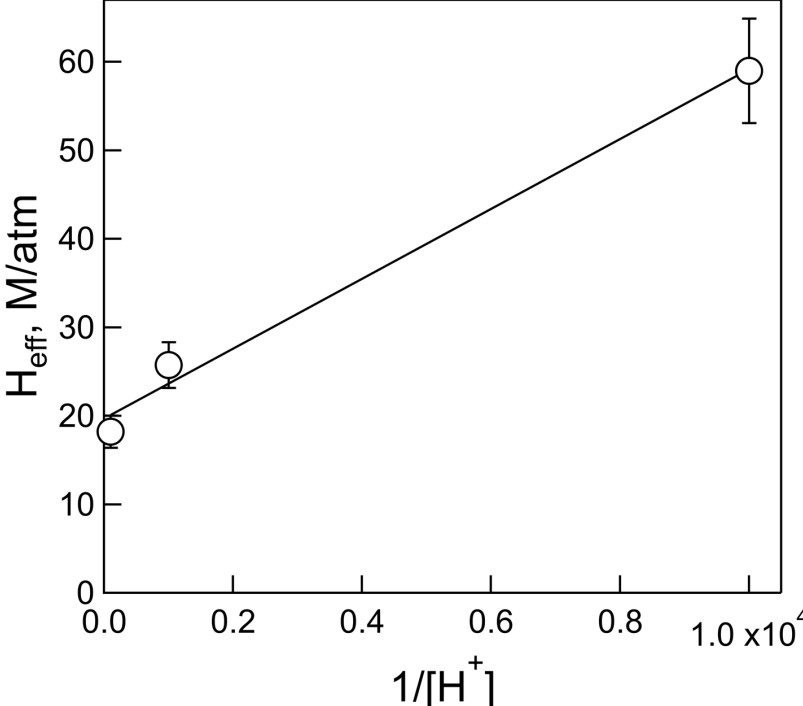

Figure 3. Plot of effective Henry's coefficient of HNCO vs 1/[H+] for the measurements at pH=2, pH=3 and pH=4,
and 298 K.

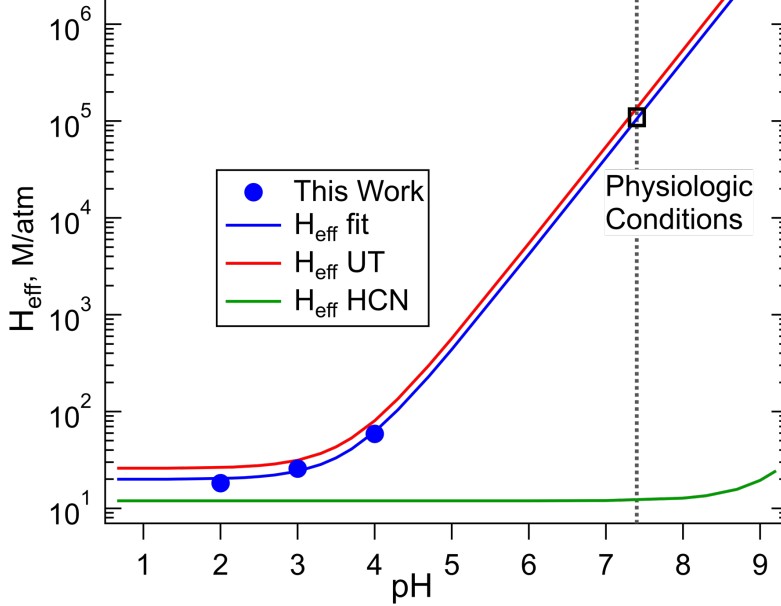

Figure 4. Comparison of effective Henry's coefficients of HNCO measured in this work (blue) with those reported
by Borduas et al., 2016, plotted versus pH, according to Equation 4. The error bars on our $H_{eff}$ values are smaller
than the width of the symbols. The green line was calculated for HCN from the intrinsic H coefficient reported by
Sander, (2015), and its pKa, 9.3.

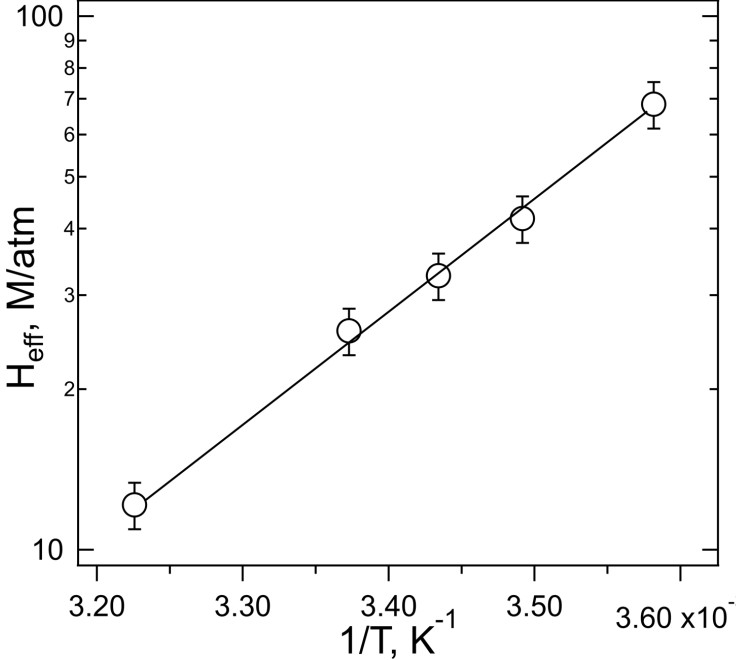

Figure 5. The plot of ln $H_{eff}$ vs 1/T for the experiments performed with HNCO at pH=3. $R^2$=0.997

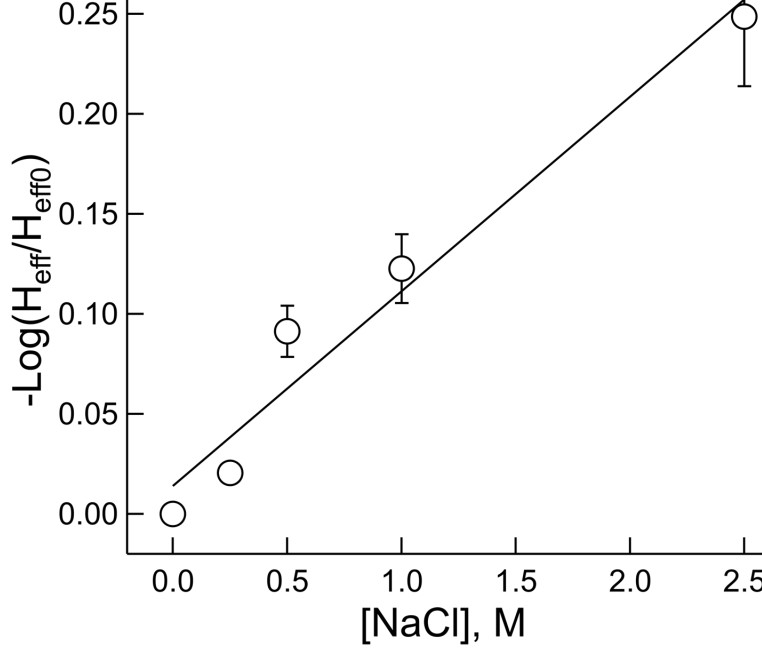

Figure 6. Dependence of the effective Henry's coefficient (H) at a given salt concentration, relative to that with no
added salt ($H_{eff0}$) versus NaCl molarity. $R^2$ = 0.960

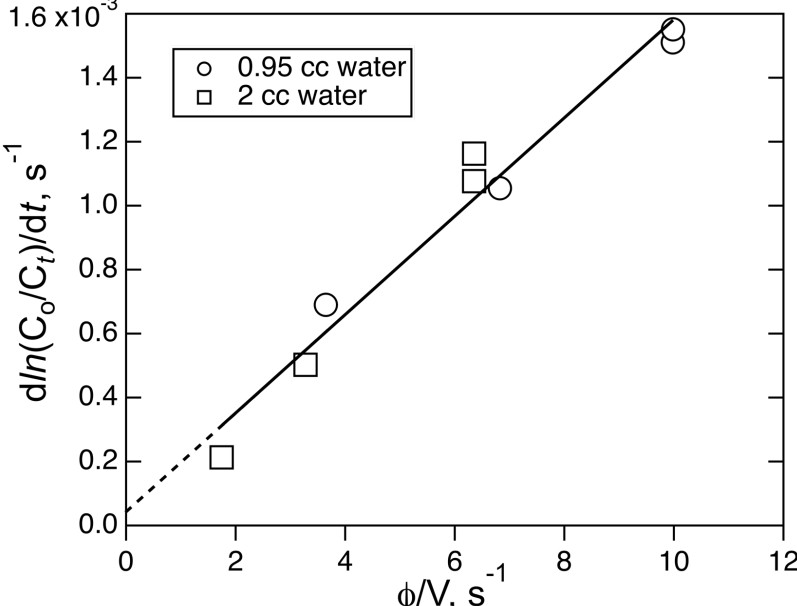

Figure 7., The Plot of ICN loss rate versus the ratio of volumetric flow rate, $\phi$, to solution volume, V, for the
experiment involving the solubility of ICN in water with the small reactor. The line is the least-square fit to the data
($R^2 = 0.968$)

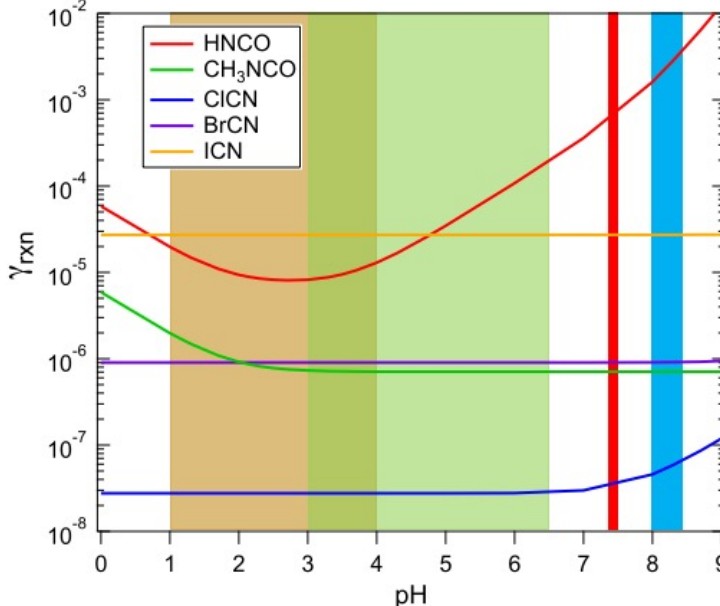

Figure 8. The uptake coefficients of HNCO, CH₃NCO, ClCN, BrCN, and ICN as a function of pH for aqueous
solution at approximately 298 K. The shaded areas show the range of pHs characteristic of: aerosols (light brown),
cloud/fog water (green), human physiology (red), and ocean surface water (light blue).

**Table 1. Summary of solubility and loss rate measurements of HNCO and CH₃NCO.**

| Solute | Solvent | Temp. (°K) | pH | Salt, Reactant | $H_{eff}$, M/atm | Literature H | $k^I$, (x10³), s⁻¹ | Literature k, (x10³), s⁻¹ | |
|---|---|---|---|---|---|---|---|---|---|
| | | | | | | | | | |
| HNCO | H₂O | 279 | 3.0 | | 68 ±7 | 73[a] | 0.22 | 0.24[a] | 0.17[b] |
| | | 286.5 | 3.0 | | 42 ±4 | 51[a] | 0.38 | 0.43[a] | 0.41[b] |
| | | 291 | 3.0 | | 33 ±3 | 40[a] | 0.66 | 0.63[a] | 0.72[b] |
| | | 296.5 | 3.0 | | 26 ±2.6 | 31[a] | 1.02 | 0.96[a] | 1.32[b] |
| | | 310 | 3.0 | | 12 ±1.2 | 17[a] | 4.15 | 2.6[a] | 5.6[b] |
| | | 298 | 2.0 | | 18 ±1.8 | | 2.2 ±0.1 | | |
| | | 298 | 3.0 | | 26 ±2.6 | | 1.02 ±0.13 | | |
| | | 298 | 4.0 | | 59 ±5.9 | | 0.72 ±0.11 | | |
| | | 298 | 3.0 | 0M NaCl | 26 ±2.6 | | | | |
| | | 298 | 3.0 | 0.25M NaCl | 24.6 ±2.5 | | | | |
| | | 298 | 3.0 | 0.5M NaCl | 20.9 ±2.1 | | | | |
| | | 298 | 3.0 | 1.0M NaCl | 19.4 ±2.0 | | | | |
| | | 298 | 3.0 | 2.5M NaCl | 14.5 ±1.5 | | | | |
| | | 292 | 3.0 | 0.45M NH₄Cl | 31.5 ±3.2 | | 1.2 | 0.005 - 0.015[c] | |
| | Tridecane (TD) | 298 | - | | 1.7 ±0.17 | | <0.043 | | |
| | TD + 10% n-Octanol | 283 | - | | 13.2 ±1.6 | | 0.16 ±0.18 | | |
| | TD + 10% n-Octanol | 298 | - | | 8.3 ±0.8 | | <0.03 | | |
| | n-Octanol | 298 | - | | 87 ±9 | | <0.015 | | |
| | n-Octanol | 310 | - | | 51 ±5 | | 0.057 ±0.014 | | |
| | | | | | | | | | |
| CH₃NCO | H₂O | 298 | 2.0 | | 1.3 ±0.13 | | 3.2 ±0.3 | 2.5[d], 3.1[e] | |
| | | 298 | 7.0 | | 1.4 ±0.14 | | 1.9 ±0.6 | 1.34[d], 1.47[e] | |
| | n-Octanol | 298 | - | | 4.0 ±0.5 | | 2.5 ±0.5 | | |
| | n-Octanol | 310 | - | | 2.8 ±0.3 | | 5.3 ±0.7 | | |
| | | | | | | | | | |

a. Calculated from the temperature and pH dependent data reported by Borduas et al., (2016).
b. Calculated from the temperature and pH dependent data reported by Jensen, (1958).
c. These were calculated from rates measured at higher pHs, assuming the mechanism is $HNCO + NH_3 => H_2NC(O)NH_2$
d. From $k_{H+}$ and $k_w$ reported by Williams and Jencks (1974a)
e. From $k_{H+}$ for HCl and $k_w$ reported by Castro et al., (1985)

**Table 2. Summary of solubility and loss rate measurements of XCN compounds.**

| Solute | Solvent | Temp. (°K) | pH | $H_{eff}$, M/atm | Literature H | $k^I$, (x10³), s⁻¹ | Literature k, (x10³), s⁻¹ |
|---|---|---|---|---|---|---|---|
| | | | | | | | |
| ClCN | $H_2O$ | 299.5 | 7.0 | 1.4 ±0.14 | 0.6[a], 0.52[b] | 0.0 ±0.42 | 3.03 x10⁻³[c] |
| | $H_2O$ | 273.15 | 7.0 | 4.5 ±0.4 | | 0.015 ±0.016 | |
| | n-Octanol | 299.5 | | 1.9 ±0.2 | | 1.3 ±0.4 | |
| | | | | | | | |
| BrCN | $H_2O$ | 296 | 7.0 | 8.2 ±0.8 | | 6.2 ±3.7 x10⁻² | 1.9 – 9.2 x10⁻²[d] |
| | $H_2O$ | 273.15 | 7.0 | 32.7 ±3 | | 2.4 ±0.5 x10⁻² | |
| | n-Octanol | 297 | | 31 ±3 | | 9 ±2 x10⁻² | |
| | | | | | | | |
| ICN | $H_2O$ | 296 | 7.0 | 270 ±54 | | 4.4 ±7.6 x10⁻² | ~3.4 x10⁻²[e] |
| | | | | | | | |

a. Measured value at 293K, reported by Weng et al., (2011).
b. Modelled value at 298K, reported by Hilal et al., (2008).
c. From $k_w$ and $k_{OH}$ reported by Bailey and Bishop (1973).
d. Estimated from Heller-Grossman et al., 1999, and Vanelslander, et al., 2012.
e. Estimated from Gerritsen et al., (1993).

**Table 3. Estimates of HNCO, CH₃NCO, and XCN compounds against loss due to heterogeneous processes.**

| Process | HNCO | CH₃NCO | ClCN | BrCN | ICN |
|---|---|---|---|---|---|
| BL deposition | 1-2 days | 0.5 yrs | yrs | 0.5 yrs | 5-10 days |
| Aerosol dep. | 6-12 days | 2-4 months | yrs | 0.6 yrs | 8 days |
| In-cloud rxn | 2-6 hrs[a] | 2 months | 10-20 wks | 1-3 wks | 1-3 days |

a. from the highly polluted case described by Barth et al., (2013).

**Table 4. Octanol/Water partition coefficients for HNCO, CH₃NCO, ClCN, BrCN and related compounds.**

| Compound | Temperature | LogK$_{ow}$ |
|---|---|---|
| | | |
| HNCO[a] | 298 | 0.64 |
| | 310 | 0.63 |
| CH$_3$NCO | 298 | 0.49 |
| ClCN | 299.5 | 0.13 |
| BrCN | 297 | 0.61[b] |
| | | |
| HC(O)OH | 298 | -0.54[c] |
| CH$_3$NO$_2$ | 293 | -0.33[c] |
| HCN | ? | 0.66[d] |
| CH$_3$CN | 298 | -0.34[c] |

a. This uses the intrinsic H calculated from Eq(5), and our results.
b. based on extrapolated H$_{H2O}$ at 297K
c. (Sangster, 1989)
d. (EPA, 1989)