# Peer review of "Solubility and Solution-phase Chemistry of Isocyanic Acid, Methyl Isocyanate, 1 and Cyanogen Halides 2 3 4 5 6 James M. Roberts1, and Yong Liu2 7 8 9 1. NOAA/ESRL Chemical Sciences Division, Boulder, Colorado, 80305 2. Department of Chemistry, University of Colorado, Denver, De"

_Atmospheric Chemistry and Physics, 2018_

## Author Comment (AC1) · 14 Nov 2018

[revised manuscript text omitted]
                                                             | I3NCO:                              |  |  |  |  |
| 489 |                                                                                                                                                      |                                     |  |  |  |  |
| 490 | $XCN + A^{-} < = > [XCN \cdot A]^{-}$                                                                                                                | (R14)                               |  |  |  |  |
| 491 | $XCN + 2H_2O \Longrightarrow HOCN + H_3O^+ + X^-$                                                                                                    | (R6)                                |  |  |  |  |
| 492 | $XCN + OH^- + H_2O => HOCN + X^- + H_2O$                                                                                                             | (R7)                                |  |  |  |  |

| 493 |                                                                                                                                                                                                                                                                                                                                                                                                                                                                                                                                                                                                                                                                                                                                                                                                                                                                                                                                                                                                                                                                                                                                                                                                                                                                                                                                                                                                                                                                                                                                                                                                                                                                                                                                                                                                                                                                                                                                                                                                                                                                                                                                               |
|-----|-----------------------------------------------------------------------------------------------------------------------------------------------------------------------------------------------------------------------------------------------------------------------------------------------------------------------------------------------------------------------------------------------------------------------------------------------------------------------------------------------------------------------------------------------------------------------------------------------------------------------------------------------------------------------------------------------------------------------------------------------------------------------------------------------------------------------------------------------------------------------------------------------------------------------------------------------------------------------------------------------------------------------------------------------------------------------------------------------------------------------------------------------------------------------------------------------------------------------------------------------------------------------------------------------------------------------------------------------------------------------------------------------------------------------------------------------------------------------------------------------------------------------------------------------------------------------------------------------------------------------------------------------------------------------------------------------------------------------------------------------------------------------------------------------------------------------------------------------------------------------------------------------------------------------------------------------------------------------------------------------------------------------------------------------------------------------------------------------------------------------------------------------|
| 494 | This complexation can be ignored in our study for CICN and BrCN since the experiment was performed in                                                                                                                                                                                                                                                                                                                                                                                                                                                                                                                                                                                                                                                                                                                                                                                                                                                                                                                                                                                                                                                                                                                                                                                                                                                                                                                                                                                                                                                                                                                                                                                                                                                                                                                                                                                                                                                                                                                                                                                                                                         |
| 495 | DI water. Accordingly, the expression for the CICN and BrCN hydrolysis rate constant is;                                                                                                                                                                                                                                                                                                                                                                                                                                                                                                                                                                                                                                                                                                                                                                                                                                                                                                                                                                                                                                                                                                                                                                                                                                                                                                                                                                                                                                                                                                                                                                                                                                                                                                                                                                                                                                                                                                                                                                                                                                                      |
| 496 |                                                                                                                                                                                                                                                                                                                                                                                                                                                                                                                                                                                                                                                                                                                                                                                                                                                                                                                                                                                                                                                                                                                                                                                                                                                                                                                                                                                                                                                                                                                                                                                                                                                                                                                                                                                                                                                                                                                                                                                                                                                                                                                                               |

[revised manuscript text omitted]

- 1059

---

## Referee Comment (RC1) · Anonymous Referee #1 · 11 Dec 2018

General comments: The manuscript presents necessary thermodynamic data for relevant atmospheric compounds including HNCO, CH3NCO, ClCN, BrCN and ICN. Specifically, values for solubility in water, octanol and tridecane and values for hydrolysis rates were determined with some pH and temperature dependence. In addition, the information on HNCO's solubility and hydrolysis are extended here to salt effect and organic solvent partitioning, useful parameters for fate modeling. The authors use established methods previously described by themselves and others. The HNCO data is well compared with existing literature. The output of this measured thermodynamic data is then used to estimate the lifetimes of HNCO, CH3NCO, ClCN, BrCN and ICN in the atmosphere against deposition, particle uptake and in cloud reactions/hydrolysis.

[Figure]

These values are important for the atmospheric chemistry modelling community and thus this manuscript is certainly appropriate for publication in ACP. Finally, the study also opens the door for further work on XCN in general and its presence in our atmosphere. I suspect the community will be prompted by this study to go and measure these compounds in ambient air.

The authors make a great point on line 196-197 that XCN could be a precursor to HNCO. I think this point links the species together very well and should be better emphasized. This important connection can be mentioned in the abstract as well as in the introduction. It is of importance to others studying the atmospheric fate of HNCO. For instance, the authors are encouraged to comment on this pathway being relevant to coastal HNCO measurements such as were made by (Zhao et al., 2014).

To further improve this manuscript, additional attention to detail is required along with presenting all experimental data, whether it is in the text or in the supplementary information. Unfortunately, only the data for HNCO partitioning is depicted, and the rest of the data is simply missing. It is necessary to include all data acquired and used to determine the experimental values listed in Table 1.

Furthermore, the organization and flow of ideas could be streamlined to be more precise and concise. Rather than organize the discussion based on compound, the discussion could be organized based on thermodynamic value. This flow would improve the readability of the manuscript, the organization of the ideas as well as the comparability between thermodynamic data among the compounds studied. Moreover, when comparing results, a hypothesis can be presented to offer an explanation as to why for example ClCN and BrCN have different solubilities in octonol (lines 548-549).

Finally, a lack of attention to formatting and quantitative detail makes this manuscript somewhat a little harder to read and follow than it should be. For instance, Table 1 is difficult to navigate, Table 2 has inconsistent units and extraneous periods, and Table 3 is missing units. IS units should be used for seconds (s rather than sec).

Principle criteria: -Scientific significance: good -Scientific quality: good/fair – can be easily improved by adding the missing data -Presentation quality: good/fair – can be improved by giving context for studying these specific compounds, streamlining the discussion, being attentive to details and adding clarity to Figures and Tables.

Reviewer recommendation: accept with revisions

Specific comments:

Abstract: In general, the abstract can be edited for conciseness: present (1) solubility rates (2) pH (3) organic solvents.

State rational for studying those specific 5 compounds. Some have previously been studied and others have not. It would be interesting to understand why these chemicals were selected.

Line 57: specify reaction rates with water

Line 61: specify which "other small nitrogen-containing compounds"

Justify the use of octonol and tridecane (although octonol is evident for Kow values, but tridecane, I am less familiar with and would like to see a brief justification and relevance to the atmosphere)

Missing concluding statement

Introduction:

First paragraph is missing references and context. What is already known about (1) the presence of these compounds in the atmosphere (2) their toxicity/ecosystem impact (3) current and gaps in knowledge relating to their atmospheric fate.

Line 177: unclear statement about electronegativity. Clarify the link between slow OH/Cl reaction rates and electronegativity

Line 205: specify range of pH and temperatures (in general try to be more quantitative)

Methods: The method is reliable and well explained. The technique does not require calibration since the authors observe a signal decay relative to a starting concentration. However, the authors give concentration ranges of their prepared standards and thus need to explain how these values were calibrated. This information could be included in the supplementary information but needs to be explained. Examples include line 231 (3% of siloxane – 3% of signal intensity? By mass?); line 238 (1% level impurity); line 235 (how was 10 ppmv mixing ratio quantified?); line 259 and so on.

Line 277-278: incorrect statement because a C-H bond (413 kJ/mol) is stronger than a N-H bond (391 kJ/mol). It is also not clear what point is being made. This discussion could benefit from being revisited.

Was CIMS used (lines 260 for instance)? (and PTRMS in line 237?) If it was, then the details of its operation should be included.

Results and discussion:

Biggest issue: all data must be shown either in the text or in the supplementary information.

Figures 3, 4 & 5: missing error bars

Line 394: specify small organic compounds

Table 1 is difficult to navigate. Merged cells could help, perhaps dividing the info into one table per compound since some columns are not necessary for all compounds. Perhaps rates can be presented in one table and thermodynamic data in another?

Lines 400-405: good discussion, but could benefit from reporting the quantitative data within the text.

Line 408: confusing "active hydrogen" terminology for an atmospheric chemist.

Line 409: an addition reaction likely occurs at the C center.

Hydrolysis rates R4 and R5 for CH3NCO are unclear to me. Does the hydrolysis go through a carbamic acid group (CH3-N(CO)-OH)? Does this group then have to be hydrolyzed with a subsequent water molecule?

Order of reaction numbering needs to be revisited to match the order the reactions were introduced.

There are also errors with the hydrolysis equations for XCN. For a hydrolysis reaction to occur, H2O cannot be on the same side of the equation. So I think R7 should read XCN + OH- → HOCN X-. R8 is a tautomerization reaction and is therefore denoted with a doubled headed arrow ↔. Tautomerization does not require H+. R8 should read HOCN ↔ HNCO.

Brief discussion on anion complexation for XCN was unclear. Do the authors therefore expect a salting in/out effect on the solubility of these compounds then?

Missing data for lines 531-534

ICN discussion missing in paragraph starting at line 546

Atmospheric and environmental chemistry implication:

Figure 8: lines 601-602 described that data from other studies are presented, but it is unclear in Figure 8 who's data is which.

Lines 631-632: knowing that (Barth et al., 2013) data used formic acid, the authors can actually specific how their own revised values could affect their modeled results.

Technical comments: Line 60: attention to significant figures in reporting Ka.

Line 64: specify the counter ion of NH4+

The SI unit for seconds is "s", not "sec", and should be corrected throughout.

Line 65: missing verb in second clause.

Check syntax of lines 105-109. Best to attribute each reference with its relevant statement here.

Arrows for all reactions should be including using symbols, like →

Line 125: define pKas

Many references are based on personal communication, and I believe that in some circumstances work/reviews can be referenced instead. For instance, indoor surfaces with chlorine to substantiate "J. Abbatt personal communication": (Wong et al., 2017)

Line 204: delete on iteration of "at several"

Line 244: specify the IUPAC name for Chloramine-T

Lines 293: already been said, could delete for conciseness.

Lines 330-331: it sounds like the manufacturer specifications had a slight temperature dependence?

Lines 339-346: repetitive

Line 567: should read "common"

Line 699: quantify "fairly readily"

Be consistent in using chemical names vs formula. (CH3CN instead of acetonitrile for instance in line 577

---

## Referee Comment (RC2) · Anonymous Referee #2 · 18 Dec 2018

Review for Roberts and Liu, 2018

This manuscript by Roberts and Liu presents a fundamental laboratory experiment to determine thermodynamic data required to predict atmospheric fates of HNCO, CH3NCO, and three XCN species. HNCO and CH3NCO are toxic volatile organic compounds for which an accurate understanding of their atmospheric fate is critical. In particular, ambient measurement of HNCO was first made possible a few years ago by Roberts and coworkers themselves. Although a number of studies have focused on the chemical behavior of HNCO since then, fundamental thermodynamic data, such as those presented here, are still lacking. XCN species are novel species whose atmospheric importance has been implied but has not been fully established.

The experiments were conducted with well established methods, measurements and analyses were performed with cautions, and the choice of experimental conditions is thoughtful. Publication in ACP should be considered, but not before substantial revisions are made.

Major comments:

1) The use of Henry's law constant.

Throughout the entire manuscript, I am concerned about the current use of effective Henry's law constant ($H_{eff}$) vs the intrinsic Henry's law constant ($H_{HNCO}$ or H). It seems that the authors fully understand the difference between the two, though the usage of H and $H_{eff}$ is inconsistent and misleading. I would recommend the authors first use a few sentences in the introduction or method to clarify the difference between $H_{eff}$ and H, and then revisit each H and $H_{eff}$ throughout the manuscript to revise them accordingly.

Here are some particular examples:

- Line 309 and Eq. (1). My understanding is that the H determined using the experimental method and Eq. (1) is in fact $H_{eff}$. The authors should clarify that.
- The only intrinsic Henry's law constant appears is in Eq. (4) and related discussions. The authors decide to temporarily use $H_{HNCO}$ here.

2) Atmospheric implication (Section IV) is one of the most important sections in the manuscript and requires some revisions. In particular:

- While the focus of this study is heterogeneous processes, the authors mention about the gas-phase fates of HNCO and CH3NCO in a rather sporadic manner. I was under an impression that the gas-phase loss of HNCO and CH3NCO is less important than the heterogeneous process, until I saw the OH rate coefficient of $CH_3NCO$ (3.6e-12 cm3 molec-1 sec-1) and realized that it is actually very important for $CH_3NCO$. I would suggest the authors extend the discussion of atmospheric fate to include gas-phase loss processes for a more complete picture.

- The introduction to the loss processes, e.g., deposition velocity, uptake coefficient, etc., is very insightful and resourceful. However, the authors perform the actual analysis at a rather abstract level after a full-bodied introduction. In particular, the fate of HNCO is summarized into a couple of numbers in Table 2, which in a sense self-negates all the detailed analyses performed by authors themselves. Given all the HNCO data at different pH and temperatures etc., HNCO

deserves a more detailed discussion in a separate paragraph, and perhaps with additional diagrams.

- It is surprising that the in-cloud rxn value of HNCO in Table 2 is directly taken from another study. Why don't the authors derive this value from their own data from this study using Eq. 16? I thought that was the whole purpose of doing all the analyses for HNCO.

- HNCO and other compounds' water solubility varies significantly across temperatures and pH. What condition is used to derive logKow in Table 2? No explanation is provided. When the Henry's law constant of HNCO varies to such an extent, is Kow of HNCO helpful at all?

3) Miscellaneous typos, mistakes, etc. Each of them is minor by itself, but the overall quality of this manuscript should be improved to achieve a professional level.

Minor and technical comments:

- Line 204 "at several at several"
- Line 284: as the authors point out, the selectivity of the detection method is indeed important. Did the authors try using CIMS and PTRMS which should be able to verify the selectivity of the Nr method?
- Line 314 "phi/V"
- Line 324 "cc/min" should be made consistent with "ml/min" used previously (Line 297)
- Line 345 redundant
- Line 348 "volumetric flow rate to solution volume" is already defined as phi/V previously.
- Line 351 "lass rate"
- Line 365 should define effective Henry's law coefficient as $H_{eff}$ here.
- Line 386 394. Please consider citing this paper for salting in/out and Setschenow constants: Wang et al. EST 2014 10.1021/es5035602
- Line 411 R10: out of curiosity, $H_2O$ can be technically treated as a type of ROH. Any suggestion on why the reaction mechanisms of HNCO towards $H_2O$ and ROH are different?
- Line 624-625: By using an extremely polluted condition, I guess the authors are trying to derive the lower limit of Aerosol Dep. lifetime. This should be clarified somewhere, perhaps as a notes to Table 2.
- Line 682 check the unit of k
- Figure 4: The figure contains data for Heff HCN, but nothing is mentioned in the caption.

---

## Author Response (AR1)

Response to Reviewers Comments
We thank the reviewers for their thorough and considered comments. In the following, the authors' responses are shown in *italics* and any added text is shown in red and the location noted by line number in the revised text.

Anonymous Referee #1

General comments: The manuscript presents necessary thermodynamic data for relevant atmospheric compounds including HNCO, $CH_3NCO$, ClCN, BrCN and ICN. Specifically, values for solubility in water, octonol and tridecane and values for hydrolysis rates were determined with some pH and temperature dependence. In addition, the information on HNCO's solubility and hydrolysis are extended here to salt effect and organic solvent partitioning, useful parameters for fate modeling. The authors use established methods previously described by themselves and others. The HNCO data is well compared with existing literature. The output of this measured thermodynamic data is then used to estimate the lifetimes of HNCO, $CH_3NCO$, ClCN, BrCN and ICN in the atmosphere against deposition, particle uptake and in cloud reactions/hydrolysis.

These values are important for the atmospheric chemistry modelling community and thus this manuscript is certainly appropriate for publication in ACP. Finally, the study also opens the door for further work on XCN in general and its presence in our atmosphere. I suspect the community will be prompted by this study to go and measure these compounds in ambient air.

*We thank the review for these kind introductory comments.*

The authors make a great point on line 196-197 that XCN could be a precursor to HNCO. I think this point links the species together very well and should be better emphasized. This important connection can be mentioned in the abstract as well as in the introduction. It is of importance to others studying the atmospheric fate of HNCO. For instance, the authors are encouraged to comment on this pathway being relevant to coastal HNCO measurements such as were made by (Zhao et al., 2014).

*We now mention this connection in the abstract with addition of the clause:*
… like the XCN species, have the potential to be a significant condensed-phase source of $NCO^-$ and therefore HNCO. (*Lines 58-59).*

*As the authors note, we have mentioned this aspect of XCN chemistry in the Introduction, and have now added a sentence at the end of Section IV.C:* Depending on the mechanism of condensed phase XCN reactions, this chemistry could be a condensed phase source of $NCO^-$ and therefore HNCO similar that observed by Zhao et al. (2014) in coastal clouds. (Lines 794-796). *We do not go any further on this point as there are no XCN measurements at mid-latitude with which to work.*

To further improve this manuscript, additional attention to detail is required along with presenting all experimental data, whether it is in the text or in the supplementary information.

Unfortunately, only the data for HNCO partitioning is depicted, and the rest of the data is simply missing. It is necessary to include all data acquired and used to determine the experimental values listed in Table 1.

*This request is contrary to accepted practice in the presentation of laboratory (or field) results, as the presentation of all the data would make this paper unnecessarily long and difficult to read. However, we are sensitive to this reviewer's desire to see more of the data that underlie the quantities presented in the paper. As a compromise, we have included more plots of the kind shown in Figures 1 and 2 in the original paper. Those are now Figures S2-S12, and represent a reasonable sampling of results for each of the compounds studied here, and include the liquid volumes and flow rates used. In addition, the raw and processed data are available on request as was made clear in the original submission.*

Furthermore, the organization and flow of ideas could be streamlined to be more precise and concise. Rather than organize the discussion based on compound, the discussion could be organized based on thermodynamic value. This flow would improve the readability of the manuscript, the organization of the ideas as well as the comparability between thermodynamic data among the compounds studied.

*We have carefully considered the reviewer's suggestion for reorganizing the paper and feel strongly that it would be best to leave it organized the way it is. Organizing by thermodynamic quantity, and then discussing each compound in turn, would be very cumbersome, since many of the thermodynamic quantities were only measured for HNCO.*

Moreover, when comparing results, a hypothesis can be presented to offer an explanation as to why for example ClCN and BrCN have different solubilities in octonol (lines 548-549).

*We have reconsidered our statement on lines 548-549 (original MS) because the relative differences between water and n-octanol solubilities of ClCN and BrCN are both about a factor of 4. This aspect is now commented on and we include a hypothesis for the different n-octanol solubilities of ClCN and BrCN. We now add the sentence:* Cyanogen chloride and BrCN have about the same relative differences in solubility in n-octanol (a factor of 3-4) as they did $H_2O$. The higher solubility of BrCN relative to ClCN could again be due to its higher dipole moment and polarizability (Maroulis and Pouchan, 1997), Lines (596-599)

Finally, a lack of attention to formatting and quantitative detail makes this manuscript somewhat a little harder to read and follow than it should be. For instance, Table 1 is difficult to navigate, Table 2 has inconsistent units and extraneous periods, and Table 3 is missing units. IS units should be used for seconds (s rather than sec).

*We now have divided up Table 1 into two tables, one for HNCO and $CH_3NCO$, and one for the XCN compounds. We hope this will make these easier to read. We have eliminated extraneous periods in what is now Table 3, but wish to keep the units the*

*way they are since hours, days, weeks and years are easier to conceptualize than if we made each quantity one unit, for example how does one convert 'years' into another unit that is easy to understand? The Table that has $K_{ow}$ data is now Table 4, perhaps the reviewer did not understand that $K_{ow}$ and therefore $LogK_{ow}$ is a unitless quantity. The time units have now been changed to the SI unit 's'*

Principle criteria: -Scientific significance: good -Scientific quality: good/fair – can be easily improved by adding the missing data -Presentation quality: good/fair – can be improved by giving context for studying these specific compounds, streamlining the discussion, being attentive to details and adding clarity to Figures and Tables.
Reviewer recommendation: accept with revisions

Specific comments:
Abstract: In general, the abstract can be edited for conciseness: present (1) solubility rates (2) pH (3) organic solvents.
*We have studied the abstract with an eye towards making it more concise, and have done so chiefly by eliminating extraneous explanation (see crossed out words and phrases). We are not sure what a "solubility rate" is, but once again we feel strongly that it is best to summarize the results by compound, and not as the reviewer has suggested. The chief reason being that there were more features of HNCO studied than the other compounds, so arranging by quantity measured, i.e. solubility, reaction rate, pH effects, organic solvents, etc. makes the presentation much more awkward and less concise.*

State rational for studying those specific 5 compounds. Some have previously been studied and others have not. It would be interesting to understand why these chemicals were selected.
*We have now added a sentence in the abstract explaining the reasons (Lines 56-59):* These nitrogen species are of emerging interest in the atmosphere as they have either biomass burning sources, i.e. HNCO and $CH_3NCO$, or like the XCN species, have the potential to be a significant condensed-phase source of $NCO^-$ and therefore HNCO.

Line 57: specify reaction rates with water
*This suggestion is not appropriate as this clause was meant to encompass the entire study and several other solvents were studied, i.e. n-octanol and tridecane.*

Line 61: specify which "other small nitrogen-containing compounds"
*We have added the phrase (Line 64);* such as HCN, acetonitrile ($CH_3CN$), and nitromethane

Justify the use of octonol and tridecane (although octonol is evident for Kow values, but tridecane, I am less familiar with and would like to see a brief justification and relevance to the atmosphere). Missing concluding statement
*In the interest of concision, we have included a justification for these two solvents in the main text:* Tridecane was used because it is the heaviest n-alkane that is still a liquid at

273.15K, and it has purely non-polar character, i.e. no functional groups, so is a slightly different model for non-polar matrices. (Lines 565-566).

*We have added the concluding phrase* (Lines70-71): features that have implications for multi-phase and membrane transport of HNCO.

*This subject is elaborated on in the main text* (Lines 625-628).

Introduction:
First paragraph is missing references and context. What is already known about (1) the presence of these compounds in the atmosphere (2) their toxicity/ecosystem impact (3) current and gaps in knowledge relating to their atmospheric fate.

*The first paragraph of the introduction was meant to set the stage for the rest of the introduction, which covers almost all of the points that the reviewer feels are missing, in extensive detail. We have added references to the first paragraph to provide context for some of the statements made, and have modified the last sentence of the paragraph, which now reads*: "which are biomass burning products, and cyanogen chloride, ClCN, cyanogen bromide, BrCN, and cyanogen iodide, ICN, which could be condensed-phase sources of cyanate ion (NCO⁻) and therefore HNCO." (Lines 103-104).

Line 177: unclear statement about electronegativity. Clarify the link between slow OH/Cl reaction rates and electronegativity

*We have changed this part of the paper as the phenomenon is much better explained by the fact that the X-CN bond is quite stable, and we have now added a reference to that effect*: Radical reaction rates (OH, Cl) have not been measured at room temperatures, but are likely to be slow due to the strength of X-CN bonds (Davis and Okabe, 1968). (Lines 183-184).

Line 205: specify range of pH and temperatures (in general try to be more quantitative)

*We have added the following text:* in the range pH 2-4, temperature in the range 279-310K, and salt concentration up to 2.5M NaCl. The rate of reaction of HNCO with $NH_4^+$ was measured at pH3, to examine the importance of this reaction to atmospheric uptake of HNCO. The solubilities of HNCO in the non-polar solvents n-octanol and tridecane were also measured as a function of temperature, in the range 298-310K, and the first-order loss rate of HNCO in n-octanol was also determined. The aqueous solubility of $CH_3NCO$ was measured at several at several pHs pH 2 and7, and the solubility in n-octanol was also determined at several temperatures, 298 and 310K (Lines 207-213).

Methods: The method is reliable and well explained. The technique does not require calibration since the authors observe a signal decay relative to a starting concentration. However, the authors give concentration ranges of their prepared standards and thus need to explain how these values were calibrated. This information could be included in the supplementary information but needs to be explained.

*We point the reviewer to lines 272 – 274 of the original paper, where we explain that we determined the concentrations in the sample stream by comparison to a nitric oxide standard. In addition, we routinely confirm our conversion efficiency using a 10ppmv HCN standard mixture. We also reference Stockwell et al., 2018 where we explain how we determined conversion efficiencies. We feel no additional explanation is needed.*

Examples include line 231 (3% of siloxane – 3% of signal intensity? By mass?);
*The siloxane impurity was 3% by mole, that is now noted on line(240).*

line 238 (1% level impurity);
This was determined on a N basis, and we now reference two papers on the instrument that was used for that determination, so the sentence now reads: The source was also analyzed by an $H_3O^+$ chemical ionization mass spectrometric system (H₃O⁺ CIMS) (Koss et al., 2018; Yuan et al., 2016), which showed that it had no impurities detectable above the 1% (as N) level. (Lines 245-246).

line 235 (how was 10 ppmv mixing ratio quantified?);
*This was a commercially prepared gas standard, the clause now reads: … a* commercially-prepared *10ppmv gas-phase standard of HCN in* $N_2$ (GASCO, Oldsmar, FL),*… (Lines 254-255).*

line 259 and so on.
*We assume the reviewer is asking how we determined the mixing ratio? This was done using the NO standard as noted above.*

Line 277-278: incorrect statement because a C-H bond (413 kJ/mol) is stronger than a N-H bond (391 kJ/mol). It is also not clear what point is being made. This discussion could benefit from being revisited.
*The point being made is this. We do not have an independent measure of the conversion efficiency of* $CH_3NCO$*, so we are arguing that the thermochemistry measured by Woo and Liu indicates that* $CH_3NCO$ *should be more easily converted than HNCO. We base this on the bond energies of the weakest bonds in the molecule, which are the ones shown. The C-H bond is not the weakest bond in the CH3NCO molecule, so the reviewer's point is irrelevant. We add the phrase:* so CH₃NCO should be easily converted by the Nr catalyst. *(Line 288).*

Was CIMS used (lines 260 for instance)? (and PTRMS in line 237?) If it was, then the details of its operation should be included.
*The* $H_3O^+$ *CIMS and PTRMS are the same instrument and we feel its operation is adequately described by the two references given. We now use only* $H_3O^+$ *CIMS as a descriptor. (Lines245-246)*

Results and discussion:
Biggest issue: all data must be shown either in the text or in the supplementary information.
*As noted above, presenting every plot for every experiment would make the paper, or supplement too long. Accepted practice is to show key examples, and to have data available as part of the publication. We have greatly expanded the data that we show in the Supplement.*

Figures 3, 4 & 5: missing error bars

*We have now included error bars for Figures 3, 5, and 6, and note that the error bars on the points in Figures 4 and 7 are smaller than the width of the symbols. We note those features in the figure captions.*

Line 394: specify small organic compounds
*We have added three examples:* such as acetylene, ethane and butane. (Line 419)

Table 1 is difficult to navigate. Merged cells could help, perhaps dividing the info into one table per compound since some columns are not necessary for all compounds. Perhaps rates can be presented in one table and thermodynamic data in another?
*We have now divided Table 1 into 2 tables.*

Lines 400-405: good discussion, but could benefit from reporting the quantitative data within the text.
*We do not see the value in repeating numbers that are given in Table 1, it would make the text cumbersome and harder to digest.*

Line 408: confusing "active hydrogen" terminology for an atmospheric chemist.
*An "active hydrogen" is one attached to a O, N, or S atom, we have now added the text:* (a hydrogen attached to an O, N, or S atom) (Lines 438-439).

Line 409: an addition reaction likely occurs at the C center.
*Yes, this is almost saying the same thing, except with the additional information that the active H ends up on the N in the HNCO molecule. We give examples to make it clear, but we have added the phrase:* where the active hydrogen ends up on the N and the other moiety ends up attached to the carbon (Line 439-440).

Hydrolysis rates R4 and R5 for CH3NCO are unclear to me. Does the hydrolysis go through a carbamic acid group (CH3-N(CO)-OH)? Does this group then have to be hydrolyzed with a subsequent water molecule?
*We thank the reviewer for the question because it has prompted us to further study the literature to clarify our explanation of the $CH_3NCO$ hydrolysis chemistry. The reviewer was correct in surmising a carbamic acid is formed (with one more H on the N atom than in the reviewer's suggested structure). We now note in the Introduction that R4 and R5 are the net reactions. (Line 156). In the Results and Discussion section we note that Castro et al. imply that the aqueous chemistry starts by hydration of $CH_3NCO$ followed either by reaction with $H_3O^+$ (fast), or $H_2O$, (relatively slow), the net reactions being R(4) and R(5). We now write:* The mechanism of $CH_3NCO$ hydrolysis and other solution chemistry is discussed by (Al-Rawi and Williams, 1977; Castro et al., 1985). The hydrolysis of $CH_3NCO$ is thought to proceed first by formation of a methyl carbamic acid:

$$CH_3NCO + H_2O \leftrightarrow CH_3NHC(O)OH \qquad\qquad (R13)$$

which is analogous to the way water adds across the N=C bond of HNCO. The methyl carbamic acid then either reacts with $H_3O^+$(faster) or $H_2O$ (slower) to produce $CH_3NH_3^+$ and $CO_2$, or $CH_3NH_2$ and $CO_2$, yielding the net reactions (R4) and (R5):

$$CH_3NCO + H_3O^+ \rightarrow CH_3NH_3^+ + CO_2 \qquad\qquad (R4)$$
$$CH_3NCO + H_2O \rightarrow CH_3NH_2 + CO_2 \qquad\qquad (R5)$$

The Henry's law measurements in our work imply that if (R13) is happening, it must be to a quite minor extent, otherwise the H constant for $CH_3NCO$ would be much larger than it is. Solution-based studies of MIC in the presence of strong acid anions (Al-Rawi and Williams, 1977; Castro et al., 1985) also imply that a complex mechanism takes place, (Lines 483-499)

Order of reaction numbering needs to be revisited to match the order the reactions were introduced.
*We have insured the reaction numbering is correct. Note that several sets of reactions were first presented in the Introduction, so might appear to be mis-numbered when brought up again in the Results and Discussion.*

There are also errors with the hydrolysis equations for XCN. For a hydrolysis reaction to occur, $H_2O$ cannot be on the same side of the equation.
*We have heard some purists claim that protons don't exist as such in aqueous solution and should always be designated $H_3O^+$. However, we have no strong attachment to this convention, so we have changed R6 to read:*
$XCN + H_2O \rightarrow HOCN + H^+ + X^-$ (Line 538)

So I think R7 should read; $XCN + OH- \rightarrow HOCN\ X-$.
*Changed:*
$XCN + OH^- \rightarrow HOCN + X^-$ (Line 539)

R8 is a tautomerization reaction and is therefore denoted with a doubled headed arrow $\leftrightarrow$. Tautomerization does not require H+. R8 should read HOCN $\leftrightarrow$ HNCO.
*The reviewer is correct that R8 is a tautomerization reaction and so we have added the double arrow. However, as with almost all tautomerizations, this reaction does not happen in the gas phase at room temperature, but requires either a surface or a solution. Therefore, it is correct to have H+ on both sides of the equation because H+ adds to one side of the molecule and dissociates from the other side, the H atom does not migrate from one end of the molecule to the other. The key piece of information is that this equilibrium, by far favors HNCO, a feature covered by Belson and Strachan.*

Brief discussion on anion complexation for XCN was unclear. Do the authors therefore expect a salting in/out effect on the solubility of these compounds then?

*We reference the studies that have hypothesized anion complexation, but note that it did not apply to our studies. We have now added some text noting that this could be the subject of further work.:*
however, such complexation should be considered in future condensed phase studies of XCN compounds. (Line 542)

Missing data for lines 531-534
*Once again, we choose not to clutter the text up with numbers that can be readily seen in the Tables.*

ICN discussion missing in paragraph starting at line 546
*We were not able to measure the solubility of ICN in n-octanol with either the large or the small reactor. We now note this in the Methods section:* Attempts to measure ICN solubilities in n-octanol were not successful using either reactor. (Line 337-338).

Atmospheric and environmental chemistry implication:
Figure 8: lines 601-602 described that data from other studies are presented, but it is unclear in Figure 8 who's data is which.
*The lines noted details about what data for which compounds are used for Figure 8, we have added the temperature and fixed a typo, but we are not sure what other designations the reviewer believes are missing. The pHs designated by the shaded areas are generally understood to apply to those matrices. This sentence now reads:* The results of these calculations are shown in Figure 8 for the H measurements at 298K reported here, $k_{hydr}$ for HNCO reported by Borduas et al., (2016), $k_{hdyr}$ for $CH_3NCO$ from this work and $k_{hydr}$ for ClCN from Bailey and Bishop (1973). (Line 653).

Lines 631-632: knowing that (Barth et al., 2013) data used formic acid, the authors can actually specific how their own revised values could affect their modeled results.
*The reviewer is correct, we can now comment on the $H_{eff}$ temperature dependence used by Barth et al., (2013), and do so by including a figure in the Supplemental Material (S13) showing the difference and include the following text:* We can point out that our results yielded slightly lower $H_{eff}$ (~22%) at the lowest temperature we measured, compared to the values used by Barth et al., (2013), see Figure S13. This would result in slightly slower removal rates in the Barth et al., model in low-temperature clouds. (Lines 684-686).

Technical comments: Line 60: attention to significant figures in reporting Ka.
*The numbers reported are the error in the fit to our data, (±0.28 x10$^{-4}$ M) propagated through log10 to obtain the uncertainty the pKa. But we see the reviewer's point here, we now round 0.28 up to 0.3 and, 0.06 up to 0.1 pH units.* (Line 62, Line 390-391)

Line 64: specify the counter ion of NH4+
*We now specify that it was $NH_4Cl$;* as $NH_4Cl$, (Line 66).

The SI unit for seconds is "s", not "sec", and should be corrected throughout.
Done – please see red highlights in text.

Line 65: missing verb in second clause.
*We do not see the problem, the verb "reaction" serves for the sentence ending on that line, and "were found to be" is the verb for the sentence that starts on that line.*

Check syntax of lines 105-109. Best to attribute each reference with its relevant statement here.
*We agree that the sentence, while correct, is awkward to read. So we have made it more concise:* There are relatively few observations of HNCO in ambient air, showing "background" mixing ratios that range from 10pptv to over several ppbv depending on the nature of regional sources, and peak mixing ratios approaching a few ppbv, observed in areas impacted by local biomass burning (Line 111)

*We disagree that the sentence needs to be parsed further as this was meant to be an introductory statement. To try to specify which paper shows what in detail does not add to the paper, as specific information from those references was not used in subsequent discussion. An exception to this is work of Zhao et al., which was dealt with at the end of the paper.*

Arrows for all reactions should be including using symbols, like $\rightarrow$
*Done, please see red highlights in text.*

Line 125: define pKas
*We have now included the sentence:* The pKa is defined as the negative $\text{Log}_{10}$ of the dissociation constant of an acid, and can be thought of as the pH at which the acid and its conjugate base (in this case BH+ and B) are at the same concentration. (Lines 132-133)

Many references are based on personal communication, and I believe that in some circumstances work/reviews can be referenced instead.
*We wish it were the case that there was published work to reference here. The BrCN and ICN data were not finalized and reported to the NASA website at the time our paper was submitted, and so by agreement, could not be referenced (nor explicitly described). That is no longer the case for BrCN, so we now reference the BrCN data on NASA data portal:* NASA, 2019 (Line 175)

*There is a much more thorough measurement of the OH + $CH_3NCO$ rate constant that is considerably lower than the literature value, but unfortunately, it has not been published, so we are only able to refer to it in vague terms. We have now updated this text:* however recent work indicates that secondary chemistry may have made this rate high by a significant amount (Papanastasiou, et al., manuscript in preparation, 2019). In addition, there are likely condensed-phase reactions that are faster than the simple hydrolysis reactions consider in this work. Never-the-less, it is useful to estimate the atmospheric loss rates implied by our work, as a baseline against which future atmosphere observations can be judged, and the importance of other heterogeneous processes can be assessed. (Lines 744-748).

For instance, indoor surfaces with chlorine to substantiate "J. Abbatt personal communication": (Wong et al., 2017).
*The reviewer is incorrect in surmising that Wong et al., 2017 describe the observation of cleaning of indoor surfaces producing HNCO. This observation was made during a project that took place in the Summer of 2018, the results of which were kindly communicated to us by one of the PIs of that project, J. Abbatt. Our understanding is that this new result has yet to be published, accordingly, we make no change to this statement.*

Line 204: delete on iteration of "at several"
*Done,* (Line 212)

Line 244: specify the IUPAC name for Chloramine-T
*This renders our description unnecessary so we include the name:* N-Chloro-p-toluenesulfonamide sodium salt *and strike the description we gave,* (Line 252-253).

Lines 293: already been said, could delete for conciseness.
*Done:* (Lines 303)

Lines 330-331: it sounds like the manufacturer specifications had a slight temperature dependence?
*Yes, we state that in the sentence.*

Lines 339-346: repetitive
*We eliminate this text, and slightly modify what is now the first sentence to read:* Examples of the data generated by equilibration experiments are shown in Figures 1 and 2, which show the exponential decays for a series of gas flow rates (Figure 1) and the correlation of the decay rates versus the ratio of volumetric flow rate to solution volume (Figure 2). Numerous other examples of both decay curves and decay rate versus $\phi$/V are shown in the Supplementary Material for a range of different analytes and solutions. (Lines 358-361).

Line 567: should read "common"
*We prefer:* commonly used (Line 617).

Line 699: quantify "fairly readily"
*We have now modified this section to read:* The results of our solubility measurements indicate that ClCN will volatilize from the condensed phase fairly readily, e.g. within seconds of the application of a thin film of chlorine bleach cleaning solution, or the bubbling of air though a spa in which ClCN is dissolved. As a result, the atmospheric removal of ClCN should be considered. (Lines 764-765).

Be consistent in using chemical names vs formula. (CH3CN instead of acetonitrile for instance in line 577
*We now ascribe to the convention of using the name and the formula when the compound is first introduced and then the formula after that. The exception is when*

*starting a sentence, then we use the name. The copy editor can correct us if that is not how the journal does it.*

Reviewer 2

This manuscript by Roberts and Liu presents a fundamental laboratory experiment to determine thermodynamic data required to predict atmospheric fates of HNCO, $CH_3NCO$, and three XCN species. HNCO and $CH_3NCO$ are toxic volatile organic compounds for which an accurate understanding of their atmospheric fate is critical. In particular¨ ambient measurement of HNCO was first made possible a few years ago by Roberts and coworkers themselves. Although a number of studies have focused on the chemical behavior of HNCO since then, fundamental thermodynamic data, such as those presented here, are still lacking. XCN species are novel species whose atmospheric importance has been implied but has not been fully established.

The experiments were conducted with well-established methods, measurements and analyses were performed with cautions, and the choice of experimental conditions is thoughtful. Publication in ACP should be considered, but not before substantial revisions are made

Major comments:
1) The use of Henry's law constant.
Throughout the entire manuscript, I am concerned about the current use of effective Henry's law constant ($H_{eff}$) vs the intrinsic Henry's law constant ($H_{HNCO}$ or $H$) It seems that the authors fully understand the difference between the two, though the usage of H and H eff is inconsistent and misleading. I would recommend the authors first use a few sentences in the introduction or method to clarify the difference between H eff and H, and then revisit each H and $H_{eff}$ throughout the manuscript to revise them accordingly
Here are some particular examples:
> -Line 309 and Eq (1). My understanding is that the H determined using the experimental method and Eq (1) is in fact $H_{eff}$. The authors should clarify that.

> -The only intrinsic Henry's law constant appears is in Eq (4) and related discussions. The authors decide to temporarily use $H_{HNCO}$ here.

*We have taken the reviewer's suggestion and use the correct designation of $H_{eff}$ for HNCO in the introduction (Line 116), and have a more thorough explanation of effective Henry's coefficient in the Results section and are careful to use the proper terms in the text (see text highlighted in red) label the Figures correctly. The use of $H_{HNCO}$ is no longer necessary, so we use H in this equation. The revised section reads as follows:*
The dependence of aqueous solubility of HNCO on pH is expected given it is a weak acid;

$HNCO_g \leftrightarrow HNCO_{aq}$          $H = [HNCO]_{aq}/[HNCO]_g$          Eq. (2)
$HNCO_{aq} + H_2O_{aq} \leftrightarrow H_3O^+ + NCO^-$          $K_a = [NCO^-][H^+]/[HNCO]$          Eq. (3)

so that what is measured is the effective Henry's coefficient, $H_{eff}$, which involves the sum of all forms of HNCO in solution:

$$H_{eff} = \{[HNCO]_{aq} + [NCO^-]\}/[HNCO]_g \qquad\qquad\qquad \text{Eq. (4)}$$

Substituting for [NCO⁻] using the rearranged form of Eq(3), and using Eq(2) we get the relationship for $H_{eff}$:

$$H_{eff} = H(1 + K_a/[H^+]) \qquad\qquad\qquad\qquad\qquad \text{Eq. (5)}$$

Lines (375-387).

*We also mention that we measure $H_{eff}$ in our experiments but the distinction is only import in the case of the weak acid HNCO. The text reads:* In practice we measure the effective Henry's coefficient in our experiments, but the distinction is only important for the weak acid, HNCO, as described in the Results and Discussion section below. (Lines 328-329).

2) Atmospheric implication (Section IV)© is one of the most important sections in the manuscript and requires some revisions. In particular:

-While the focus of this study is heterogeneous processes, the authors mention about the gas-phase fates of HNCO and CH3NCO in a rather sporadic manner. I was under an impression that the gas-phase loss of HNCO and CH3NCO is less important than the heterogeneous process, until I saw the OH rate coefficient of CH3NCO (3 6e-12 cm3 molec-1 sec-1) and realized that it is actually very important for CH3NCO. I would suggest the authors extend the discussion of atmospheric fate to include gas-phase loss processes for a more complete picture.

*As described in the response to reviewer 1, this is a difficult issue for us to address, since there is a thorough study of the OH + CH₃NCO rate constant that has resulted in a significantly lower rate constant, and addresses the probable reason for the high result in the single measurement reported in the literature. Unfortunately, it has not been published yet so we cannot quote it directly. We have modified this section to note that the literature value is in question, citing a personal communication (D. Papanastasiou). Moreover, there is the real possibility that condensed phase reactions other than hydrolysis could compete with the OH reaction rate. Accordingly, it is useful to estimate the heterogeneous loss processes of CH₃NCO due to the chemistry that we currently know. We have added to this paragraph:*

The atmospheric chemistry of CH₃NCO is less well studied than HNCO. There is a single reported measurement of the reaction rate of CH₃NCO with OH by relative rates which gave $k = 3.6 \times 10^{-12}$ cm³ molec⁻¹ s⁻¹ (Lu et al., 2014), however recent work indicates that secondary chemistry may have made this rate high by a significant amount (Papanastasiou et al., manuscript in preparation, 2019). In addition, there are likely condensed-phase reactions that are faster than the simple hydrolysis reactions consider in this work. Never-the-less, it is useful to estimate the atmospheric loss rates implied by this work, as a baseline against which future atmosphere observations can be judged, and the importance of other heterogeneous processes can be assessed. (Lines 744-748)

-The introduction to the loss processes, e.g., deposition velocity, uptake coefficient, etc., is very insightful and resourceful. However, the authors perform the actual analysis at a rather abstract level after a full-bodied introduction. In particular, the fate of HNCO is summarized into a couple of numbers in Table 2, which in a sense self-negates all the detailed analyses performed by authors themselves. Given all the HNCO data at different pH and temperatures etc., HNCO deserves a more detailed discussion in a separate paragraph, and perhaps with additional diagrams.

*We agree that our results allow for a better description of HNCO loss processes compared to the previous studies, e.g. (Barth et al., 2013), however a detailed analysis is beyond the scope of this paper. Simple calculations don't allow us to refine the highly polluted case modeled by Barth et al., so we prefer to use that number in our Table 3 as a limiting case. The possible impact of higher solubility in organic substrates on the aerosol loss processes is now explored briefly by estimating the lifetime against reaction for mixed organic/aqueous phase particles. This section now reads:* Given that aerosol particles in most polluted atmospheres are at least half organic carbon by mass (Jimenez and et al., 2009), it is useful to estimate what effect an increased solubility of HNCO might have on its removal lifetime. If take our 1000 $\mu m^2/cm^3$ surface area aerosol from the above calculation, assume a 50/50 organic to aqueous distribution and that the solubility of HNCO in the organic fraction is the same as n-octanol, we can arrive at a weighted average Henry's solubility of 55 M/atm. If we combine that with the same reaction rate corresponding to pH1, then the lifetime of HNCO against reaction to this aerosol drops to about 2 days, a significant effect.

(Lines 724-730).

-It is surprising that the in-cloud rxn value of HNCO in Table 2 is directly taken from another study. Why don't the authors derive this value from their own data from this study using Eq. 16? I thought that was the whole purpose of doing all the analyses for HNCO.

*We have now modified this section with the material added above, but wish to keep the number from the Barth et al., 2013, as it was calculated using a fully coupled model.*

-HNCO and other compounds' water solubility varies significantly across temperatures and pH What condition is used to derive logKow in Table 2? No explanation is provided. When the Henry's law constant of HNCO varies to such an extent, is Kow of HNCO helpful at all?

*We listed the temperatures in column 2 of the Table and we now make it clear that we use the intrinsic H coefficients calculated from our work. The reviewer is correct to wonder how $K_{ow}s$ for weak acids can have application at physiologic pH, a point we should have explained further. We have now added text and a reference to explain that membrane transport theory uses $K_{ow}s$ and $pK_as$ of weak acids to estimate transport rates. In other words, such models account for the weak acid equilibrium and the solubility differences between aqueous and membrane media. Our results show directly that HNCO is more efficiently transported by membranes than a common weak acid, formic acid. We have now included this material in the section on $K_{ow}$:* However, transport models of biological systems account for these acid base equilibria along with using the $K_{ow}$ to estimate transport rates (Missner and Pohl, 2009). Formic acid is a similarly weak acid (pKa = 3.77) and so is a good point of comparison to HNCO. The n-octanol partition coefficient of HNCO is a factor of 15 larger than that of HC(O)OH, so should have larger membrane permeabilities. (Lines 625-626).

3) Miscellaneous typos, mistakes, etc. Each of them is minor by itself, but the overall quality of this manuscript should be improved to achieve a professional level.

Minor and technical comments:
        -Line 204 "at several at several"
*This has now been fixed, see response to reviewer 1 above.*

        -Line 284: as the authors point out, the selectivity of the detection method is indeed important. Did the authors try using CIMS and PTRMS which should be able to verify the selectivity of the Nr method?
*We did not use CIMS or PTRMS to verify the selectivity of our Nr method since it is well established that the solution-phase chemistry of these compounds leads to highly soluble products that do not come out of solution under the conditions used here.*

        -Line 314 "phi/V"
*Corrected, (Line 324).*

        -Line 324 "cc/min" should be made consistent with "ml/min" used previously (Line 297)
*We prefer to use cc instead of mL, so all the units have been made consistent. Please see text noted in red throughout.*

        -Line 345 redundant
*This section was removed in changes suggested by reviewer 1.*

        -Line 348 "volumetric flow rate to solution volume" is already defined as phi/V previously
*Now changed to $\phi$/V, (Line 359).*

        -Line 351 "lass rate"
*Corrected: (Line 364).*

        -Line 365 should define effective Henry's law coefficient as $H_{eff}$ here
*As noted above, we have moved this discussion to the Introduction.*

        -Line 386 394. Please consider citing this paper for salting in/out and Setschenow constants: Wang et al. EST 2014 10.1021/es5035602
*We now add a sentence on the results of Wang et al., who found that Setschenow constants for ammonium sulfate are larger than those for NaCl:* Interestingly, Wang et al., (2014) found that Setschenow constants for ammonium sulfate {$(NH_4)_2SO_4$} are typically larger than those for NaCl, a feature which might impact the uptake of HNCO to aerosol particles having substantial $(NH_4)_2SO_4$ content. (Lines 420-422).

-Line 411 R10: out of curiosity, $H_2O$ can be technically treated as a type of ROH. Any suggestion on why the reaction mechanisms of HNCO towards $H_2O$ and ROH are different?

*This is a really good question. The initial mechanisms are the same, both HOH and ROH add across the N=C bond in the HNCO molecule. The difference is that HOH makes carbamic acid [$H_2NC(O)OH$] which is unstable and decomposes to $NH_3$ and $CO_2$, and ROHs make carbamates ($H_2NC(O)OR$, carbamic acid esters), which are stable molecules. We now have added a sentence in this section to that effect*: Note that this is really the same mechanism as the neutral hydrolysis of HNCO, except that the addition of water forms carbamic acid, $H_2NC(O)OH$ which is unstable and decomposes to $NH_3$ and $CO_2$. (Lines 445-446).

-Line 624-625: By using an extremely polluted condition, I guess the authors are trying to derive the lower limit of Aerosol Dep. Lifetime. This should be clarified somewhere, perhaps as a notes to Table 2.

*That was correct, we wished to estimate a lower limit to lifetime against aerosol deposition. We have now added a clause to that sentence: … to obtain a lower limit to the lifetime against this process,* (Line 676-677)

-Line 682 check the unit of k

*The reviewer is correct that we had the units of the rate constant wrong, that has now been corrected;* (Line 774-775).

-Figure 4: The figure contains data for $H_{eff}$ HCN, but nothing is mentioned in the caption.

[revised manuscript text omitted]

Unlike HNCO and CH3NCO, some of the XCN compounds have absorbances in the near UV-vis that could lead to photolysis in the lower atmosphere, Figure S14. The UV-vis spectra and photon fluxes estimated from the NCAR TUV model (NCAR, 2018) can be used to calculate photolysis rates, by integrating over the wavelength region where the absorption is significant, and assuming a quantum yield of 1. The absorption spectra are such that ClCN will not be photolyzed in the troposphere, BrCN has some slight absorption in the actinic region and ICN has substantial absorption. The lifetimes against photolysis at 0km altitude, 40°N, on June 30. 2015, were estimated to be 135 days for BrCN, and 9 hours for ICN.

[Figure]

Figure S14. The UV-vis absorption spectra of ClCN, BrCN, ICN, (Barts and Halpern, 1989; Felps et al., 1991; Hess and Leone, 1987; Russell et al., 1987) and the photon flux spectrum estimated from the NCAR TUV model for 40° N, surface on June 30, 2015 (NCAR, 2018). The extrapolation assumes the cross-sections are ln-linear over the portions that tail into the actinic region.

---

## Author Response (AR2)

Response to Co-Editor's Comments
We thank the co-editor for his careful reading of our revised manuscript. In the following, the authors' responses are shown in *italics* and any added or modified text is shown in red.

Comments to the Author:
Dear authors. I have the manuscript a final read a spotted a few minor things listed below.
I am somewhat concerned by the fraction of sources cited as "personal communications". I would avoid those if at all possible and verify whether the material is already published by now.
*We agree that such references should be avoided if possible. We have removed the "personal communication" reference to the BrCN observations because those are now part of the NASA Atom-3 dataset, hence are publicly available. We would like to keep the reference to ICN observations because they are interesting, but won't be reported, or otherwise published any time soon. Likewise, the observation of HNCO produced after cleaning with chlorine bleach solution is sufficiently important that it should be included and we feel it important to make sure the group that communicated it to us gets credit for it, so we'd like to keep that one and add a collaborator's name. We have also added them to the acknowledgement section.*

Minor corrections:
L64: missing + sign after NH4
L74: remove strikethrough formatting from was
L109: 10pptv -> 10 pptv
L113: by Roberts et al., (Roberts et al., 2011) -> by Roberts et al. (2011)
L113: by Borduas et al., (2016) -> Borduas et al. (2016) [and numerous other similar replacements throughout the text]
*These corrections have been made.*

L129: only at the pKas of the BH+, which are typically pH 9-10 -> only for BH+, which have pKa ranging from 9-10.
*The co-editor's suggestion changes the meaning of the sentence, and makes it incorrect. The rates were measured at the pKa of the conjugate acid, but the rate constants were calculated assuming the reaction involved the neutral species, e.g. HNCO + NH$_3$, and corrections were made for the acid-base equilibria. The phrase suggested is not appropriate, and we would like to leave it unchanged.*

L130: the sentence explaining pKa is unnecessary
*This was added at the request of one of the original reviewers, we are happy to remove it since the co-editor thinks it is not necessary.*

L155: should H2O+H+ be replaced by H3O+ to make the reaction analogous to R1?
*Yes, this is now corrected.*

L164: (see for example (Afifi and Blatchley III, 2015) -> (see for example Afifi and Blatchley III (2015))

L166, L169: - should be superscript
L168: (see for example (Heller-Grossman et al., 1999) -> (see for example Heller-Grossman et al. (1999)) [and other similar replacements]
L190: are known … and so involve -> is known … and so involves
*These are now corrected.*

R6, R8: I would use H3O+ instead of H+
*Once again, these were changed after one of the reviewers did not like the use of $H_3O^+$. We are happy to change them back.*

L210: remove strikethrough formatting from "at several"
L241: 10ppmv -> 10 ppmv [and other similar space insertions in other places in the text, e.g., L255]
L255: I would use cm3 instead of cc, here and elsewhere in the text
L267: unformatted reference here by Warneke
L367-377: please use correct arrows for the equilibrium reactions
L373: - should be superscript
L379: + should be superscript
L390: H -> Heff
L395: no need to provide formula of formic acid
410: no need to provide formula of ammonium sulfate
R11, R12, R13, R14, R15: please use correct arrows for the equilibrium reactions
L469: MIC is introduced without definition. I do not think there is need to for it, CH3NCO is not that much longer.
*These have all been changed.*

L638: please confirm that this formula should use the intrinsic (not effective) H
*In fact this formula should use $H_{eff}$, we have now corrected it.*

L642: velocity -> speed
L736: Never-the-less -> Nevertheless
L1158: + should be superscript
L1176: H -> Heff
L1180: remove comma
*These have no all been corrected.*

[revised manuscript text omitted]

Warneke, C., Trainer, M. T., de Gouw, J. A., Parrish, D. D., Fahey, D. W., Ravishankara, A. R., Middlebrook, A.
M., Brock, C. A., Roberts, J. M., Brown, S. S., Neuman, J. A., Lerner, B. M., Lack, D., Law, D., Huebler, G.,
Pollack, I., Sjostedt, S., Ryerson, T. B., Gilman, J. A., Liao, J., Holloway, J., Peischl, J., Nowak, J. B., Aikin, K.,
Min, K.-E., Washenfelder, R. A., Graus, M., Richardson, M., Markovic, M. Z., Wagner, N. L., Welti, A., Veres, P.
R., Edwards, P., Schwarz, J. P., Gordon, T., Dube, W. P., McKeen, S. A., Brioude, J., Ahmadov, R., Bougiatioti, A.,
Lin, J. J., Nenes, A., Wolfe, G. M., Hanisco, T. F., Lee, B. H., Lopez-Hilfiker, F. D., Thornton, J. A., Keutsch, F. N.,
Kaiser, J., Mao, J., and Hatch, C. D.: Instrumentation and measurement strategy for the NOAA SENEX aircraft
campaign as part of the Southeast Atmosphere Study 2013, Atmos. Meas. Tech., 9, 3063-3093, 2016.

[revised manuscript text omitted]